# ForceFM: Enhancing Protein-Ligand Predictions through Force-Guided Flow Matching

**Huanlei Guo**[1]    **Song Liu**[1]    **Bingyi Jing**[123*]

[1] Department of Statistics and Data Science, Southern University of Science and Technology
[2] School of Artificial Intelligence, The Chinese University of Hong Kong, Shenzhen
[3] Shenzhen Loop Area Institute

## Abstract

Molecular docking is a fundamental technique in structure-based drug discovery, playing a critical role in predicting the binding poses of protein-ligand complexes. While traditional docking methods are generally reliable, they are often computationally expensive. Recent deep learning (DL) approaches have substantially accelerated docking and improved prediction accuracy; however, they frequently generate conformations that lack physical plausibility due to insufficient integration of physical priors. To deal with these challenges, we propose ForceFM, a novel force-guided model that integrates a force-guided network into the generation process, steering ligand poses toward low-energy, physically realistic conformations. Force guidance also halves inference cost compared with the unguided approaches. Importantly, replacing the guiding potential with diverse energy functions-including Vina, Glide, Gnina, and Confscore-preserves or improves performance, underscoring the method's generality and robustness. These results highlight ForceFM's ability to set new standards in docking accuracy and physical consistency, surpassing the limitations of previous methods. Code is available at `https://github.com/Guhuary/ForceFM`.

## 1   Introduction

Molecular docking is a critical component of structure-based drug discovery [1], aiming to predict the predominant binding modes of protein-ligand complexes based on experimentally determined or computationally modeled protein structures and ligands. Traditional docking programs such as AutoDock 4 [2], AutoDock Vina [3], Glide [4], and GOLD [5] use heuristic search algorithms to explore a range of possible ligand conformations. These programs employ scoring functions grounded in physics and chemistry principles to estimate binding strengths and select optimal poses. Despite their effectiveness, these classical methods often involve high computational costs, leading to slow processing and significant resource demands, which limit their applicability in large-scale drug discovery projects. In high-throughput screening scenarios, researchers may need to dock millions of molecules daily. The prohibitive runtime of traditional methods thus significantly constrains their utility in practical pharmaceutical pipelines.

Motivated by the success of deep learning (DL) and geometric learning (GL) in diverse 3D generation tasks in the vision community, recent works have begun to leverage these techniques for molecular docking. This has given rise to two broad categories of DL-based docking approaches. The first category encompasses regression-based models, which directly predict the ligand coordinates in 3D space [6, 7, 8, 9]. The second category relies on generative modeling, producing multiple candidate poses for selection [10, 11, 12]. Although these methods significantly improve efficiency

---

*Corresponding authors: Bingyi Jing (jingby@sustech.edu.cn).

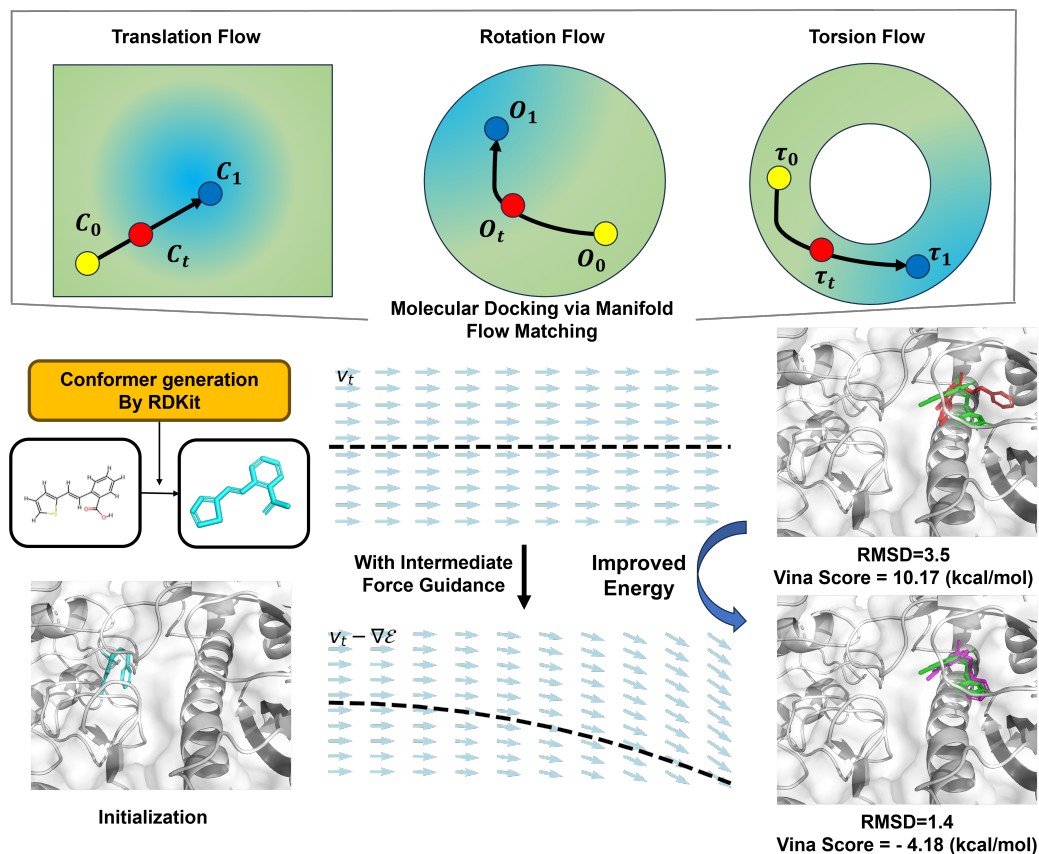

Figure 1: Overview of ForceFM. The baseline flow matching model operates on translational, rotational, and torsional degrees of freedom. By incorporating additional force guidance based on physical principles, ForceFM generates ligands with lower energy. Color scheme: **Blue**—initialization, **Green**—ground truth, **Red**—sampled ligand without force guidance, **Purple**—sampled ligand with force guidance.

and accuracy (e.g., in terms of the root-mean-square deviation, RMSD), a notable shortcoming remains: many generated conformations are physically implausible, exhibiting steric clashes or residing in high-energy regions [13], primarily because physical priors are under-utilized in the model design.

However, purely data-driven methods often fail to incorporate critical physical constraints, which can lead to unphysical poses. In real-world drug discovery, low-energy and physically consistent poses are vital. To address this gap and combine the advantages of both traditional physics-based approaches and deep generative methods, we propose a novel *force-guided flow matching* model called **ForceFM**. Specifically, ForceFM integrates physics-based priors in the form of guiding force terms derived from multiple scoring functions, including the Vina score [3], Gnina score [14], Glide score [4], and Confscore [10]. By explicitly modeling a force-guidance network with these diverse energy functions, our approach steers the generative process toward lower-energy conformations, thereby improving physical plausibility while also improving the sampling efficiency. Moreover, it demonstrates robust generalization in blind docking scenarios, showing promising performance on unseen examples. This aligns with the growing demand for generalization in DL-based molecular docking, addressing concerns about overfitting to specific protein or ligand types and broadening its applicability across diverse drug discovery tasks.

In summary, our main contributions are as follows: (1) **Innovative Docking Model.** We present a force-guided flow matching model for molecular docking that integrates deep learning with multiple energy functions (e.g., Vina, Glide, Gnina, Confscore). This hybrid approach combines the flexibility of deep learning with the physical accuracy of energy-based methods, guiding the generative process

toward low-energy conformations with fewer steps, improving computational efficiency, especially in high-throughput screening applications. (2) **Comprehensive Benchmark Validations.** Extensive experiments on the PDBBind dataset demonstrate that our model outperforms existing methods in both docking accuracy and physical plausibility, consistently generating more realistic ligand poses. Moreover, we evaluate the generalization of our approach across multiple energy functions, confirming its broader applicability in various docking scenarios.

## 2 Related Works

### 2.1 Molecular Docking

Molecular docking, or protein-ligand docking, is a computational technique for predicting the optimal binding pose of a protein-ligand complex. Traditional approaches, such as AutoDock 4 [2], AutoDock Vina [3], Glide [4], and GOLD [5], use heuristic search algorithms to explore possible ligand conformations. These methods rely on physics-based energy functions to evaluate, rank, and refine ligand structures. However, with recent advances, geometric deep learning has emerged as a promising approach for docking predictions. These approaches can be broadly categorized into regression-based and sampling-based methods.

**1) Regression-based methods** aim to directly predict ligand binding coordinates or refine structures by predicting pairwise atomic distances. Notable methods in this category include EquiBind [6], TankBind [15], E3Bind [7], FABind [8], and KarmaDock [16].

**2) Sampling-based methods** generate multiple ligand poses and then optimize or select from the sampled conformations [17, 10, 11]. Although these methods are computationally more demanding than regression-based models, they generally yield more accurate predictions.

In addition, Alphafold-latest achieves huge breakthroughs, but in close-form introduction without details. DeltaDock [18] and HelixDock [19] also obtain competitive results for us. However, they either generate large-scale data with simulators or add external high quality data for the training, which is not a fair comparison.

### 2.2 Guided Generation

In generative modeling, controllable generation is essential to align diffusion models with desired outputs. Classifier guidance [20] and classifier-free guidance [21] are prominent approaches that conditionally guide diffusion models, achieving impressive results in applications like text-to-image [22, 23] and video generation [24]. Recently, Zhou *et al.* [25] adopts an energy-based perspective to understand the confidence model. FlowAB [26] adopts the important physical prior knowledge into the flow model to guide the antibody generation, which is not an exact guidance. Lu *et al.* [27] proposed an exact energy guidance policy using a scalar reward function, rather than fixed conditioning, while Wang *et al.* [28] introduced CONFDIFF for protein conformation generation. These advances motivate our proposed exact force guidance strategy, which uses an equivariant network to approximate the intermediate force vector directly.

## 3 Proposed Method

This section describes our force-guided flow matching framework, **ForceFM**, which generates physically plausible ligand conformations by integrating physics-based priors into a data-driven manifold flow matching approach. Section 3.1 briefly reviews Riemannian Flow Matching, while Section 3.2 introduces our *baseline* manifold flow matching model that operates on the translational, rotational, and torsional degrees of freedom of ligands. Finally, Section 3.3 presents our key contribution: a *force-guided* generation strategy that incorporates the different energy function as a corrective signal to ensure low-energy conformations.

### 3.1 Preliminaries: Riemannian Flow Matching

Let $\mathcal{P}(\mathcal{M})$ represent the space of probability distributions defined on a manifold $\mathcal{M}$ endowed with a Riemannian metric $g$. Suppose we have a data distribution $q(\mathbf{x})$ and a prior $p(\mathbf{x})$. We

define a probability path $p_t : [0, 1] \to \mathcal{P}(\mathcal{M})$ interpolating between $p_0$ and $p_1$, with corresponding tangent vector fields $u_t(\mathbf{x}) \in \mathcal{T}_x\mathcal{M}$. Flow Matching (FM) [29, 30] approximates $u_t(\mathbf{x})$ by a learned vector field $v_t(\mathbf{x})$, minimizing $L_{\text{RFM}}(\theta) = \mathbb{E}_{t, p_t(\mathbf{x})}\big\|v_t(\mathbf{x}) - u_t(\mathbf{x})\big\|_g^2$, where $\| \cdot \|_g$ is the Riemannian norm and $\theta$ the parameters of $v_t$. In practice, we use the Conditional Flow Matching (CFM) formulation: $L_{\text{CRFM}}(\theta) = \mathbb{E}_{t\sim\mathcal{U}(0,1),\, p_1(\mathbf{x}_1),\, p_t(\mathbf{x}|\mathbf{x}_1)}\big\|v_t(\mathbf{x}) - u_t(\mathbf{x} \mid \mathbf{x}_1)\big\|_g^2$, which is easier to compute. Once trained, sampling from the data distribution is performed by integrating the learned ODE $\frac{d}{dt}\varphi_t(\mathbf{x}) = v_t(\varphi_t(\mathbf{x}))$, transforming points from the prior $p_0$ to $p_1$.

### 3.2 Baseline: Manifold Flow Matching

To adapt Flow Matching to ligand docking, we note that a ligand's pose can be decomposed into three main types of geometric transformations [10]: *translation*, *rotation*, and *torsion angles* (internal rotation around rotatable bonds). Each transformation lives on a distinct manifold: Translation in $\mathbb{R}^3$, Rotation in $SO(3)$ and Torsion angles in $(SO(2))^m$ (a hypertorus).

Hence, the ligand pose is represented as $\mathbf{x} \triangleq \big(C, O, \boldsymbol{\tau}_1, \ldots, \boldsymbol{\tau}_m\big)$, where $C \in \mathbb{R}^3$, $O \in SO(3)$, and $\boldsymbol{\tau}_i \in [-\pi, \pi)$. This decomposition naturally leverages the Lie group structures $\mathbb{T}(3)$, $SO(3)$, and $SO(2)$ for each respective degree of freedom, aligning well with Riemannian Flow Matching.

#### 3.2.1 Translational Degree of Freedom

To model the translation, we treat the path of the center $C_t$ as a flow on a Euclidean manifold. We use a vanilla Gaussian flow matching (CFM) approach on this manifold, which employs independent coupling techniques [31] to model the conditional flow:

$$C_t = t\, C_1 + (1 - t)\, C_0, \tag{1}$$

where $C_0 \sim \mathcal{N}(0, \sigma_{\text{tr,max}}^2 I)$ is sampled from an isotropic Gaussian (the prior), and $C_1$ is the ground-truth ligand translation. The corresponding CFM objective is

$$L_{\text{tr}}(\theta) = \mathbb{E}_{t, q_1(\mathbf{x}_1), q_0(\mathbf{x}_0)} \|v_{t,\text{tr}}(\mathbf{x}) - C_1 + C_0\|_2^2. \tag{2}$$

#### 3.2.2 Rotational Degree of Freedom

To model the probability path of rotation matrix $O_t \in SO(3)$, we adopt $SO(3)$-CFM [32] as

$$O_t = \exp_{O_0}\big(t\, \log_{O_0}(O_1)\big), \tag{3}$$

where $\exp$ and $\log$ are the exponential and logarithmic maps on $SO(3)$, which can be efficiently computed using Rodrigues' formula and the Lie algebra $\mathfrak{so}(3)$. The prior distribution to sample $O_0$ is defined as isotropic Gaussian distribution on $SO(3)$ [33]. Then the loss is

$$L_{\text{rot}}(\theta) = \mathbb{E}_{t, q_1(\mathbf{x}_1), q_0(\mathbf{x}_0)} \left\| v_{t,\text{rot}}(x_t) - \frac{\log_{O_t}(O_0)}{t} \right\|_{SO(3)}^2. \tag{4}$$

#### 3.2.3 Torsional Degree of Freedom

For a ligand with $m$ rotatable bonds, the torsion angles lie in $[-\pi, \pi)^m$. For the torus, the manifold is the quotient space $\mathbb{R}^m/2\pi\mathbb{Z}$, leading to the equivalence relations with period $2\pi$ [34]. We choose the prior distribution $p_0$ as the product of standard wrapped normal distribution as:

$$p_0(\boldsymbol{\tau}) \propto \prod_{i=1}^{N} \sum_{d\in\mathbb{Z}} \exp\left(-\frac{\big\|\boldsymbol{\tau}^{(i)} + 2\pi d\big\|^2}{2}\right). \tag{5}$$

The path $\boldsymbol{\tau}_t(\boldsymbol{\tau}_0, \boldsymbol{\tau}_1)$ is then an interpolation on the torus [34]. We approximate the geodesic distance with a linear shift to handle $2\pi$ periodicities:

$$\begin{aligned} \boldsymbol{\tau}_0' &= (\boldsymbol{\tau}_0 + \pi) \bmod (2\pi) - \pi, \\ u_t\left(\boldsymbol{\tau}_0', \boldsymbol{\tau}_1\right) &= (\boldsymbol{\tau}_1 - \boldsymbol{\tau}_0' + \pi) \bmod (2\pi) - \pi, \\ \boldsymbol{\tau}_t\left(\boldsymbol{\tau}_0', \boldsymbol{\tau}_1\right) &= \boldsymbol{\tau}_0' + t * u_t\left(\boldsymbol{\tau}_0', \boldsymbol{\tau}_1\right). \end{aligned} \tag{6}$$

The conditional Flow Matching loss is:

$$L_{\text{tor}}(\theta) = \mathbb{E}_{t,\, q_1(\mathbf{x}_1),\, q_0(\mathbf{x}_0)}\big\|v_{t,\text{tor}}(\mathbf{x}_t) - u_t(\boldsymbol{\tau}_0', \boldsymbol{\tau}_1)\big\|_2^2. \tag{7}$$

### 3.2.4 Baseline Flow Matching Objective

We combine these three losses with respective weights:

$$L(\theta) = \lambda_{\mathrm{tr}} L_{\mathrm{tr}} + \lambda_{\mathrm{rot}} L_{\mathrm{rot}} + \lambda_{\mathrm{tor}} L_{\mathrm{tor}}. \tag{8}$$

Here, we set $\lambda_{\mathrm{tr}} = 1$, $\lambda_{\mathrm{rot}} = 1$, $\lambda_{\mathrm{tor}} = 1$. After training, sampling is performed by solving the ODE $\frac{d}{dt}\mathbf{x} = v_t(\mathbf{x})$ with $\mathbf{x}_0$ drawn from the prior. *This baseline approach* is purely data-driven, relying on learned distributions of ligand poses, but it does *not* explicitly ensure low-energy or sterically valid configurations. As we will see, *unconstrained flow matching* can sometimes produce high-energy or overlapping geometries in the absence of physical priors.

### 3.3 Force-Guided Generation

To address the limitations of baseline flow matching and ensure physically valid docking poses, we introduce a *force-guided* strategy inspired by energy-based diffusion methods [27, 28]. Our goal is to bias ligand conformations toward low-energy regions of the conformational space by incorporating an *additional force term* derived from a chosen scoring function.

Let $q_1(\mathbf{x}_1)$ be the distribution generated by our baseline flow model in Section 3.2, and let $\mathcal{E}_1(\mathbf{x}_1)$ be the energy function for conformation $\mathbf{x}_1$ given protein $\mathbf{y}$ ($\mathbf{y}$ is omitted for simplification). We aim to form a Boltzmann-like distribution:

$$p_1(\mathbf{x}_1) \;\propto\; q_1(\mathbf{x}_1) \, \exp\!\big[-k\,\mathcal{E}_1(\mathbf{x}_1)\big], \tag{9}$$

where $k$ is an inverse temperature factor controlling the strength of energy guidance. By incorporating this additional energy term, $p_1(\mathbf{x}_1)$ provides a more accurate estimate and generates ligands that adhere to physical plausibility.

#### 3.3.1 Deriving the Force-Guided Flow

The following theorem suggests a force-guided flow from a conditional vector field to generate the modified probability distribution.

**Theorem 3.1** *Given an energy function $\mathcal{E}_1(\cdot)$ and a conditional flow $u_t(\mathbf{x} \mid \mathbf{x}_1)$ that generates the probability distribution $q_t(\mathbf{x} \mid \mathbf{x}_1)$. Assume the guided distribution has the form $p_t(\mathbf{x}) \propto q_t(\mathbf{x}) \exp(-k\mathcal{E}_t(\mathbf{x}))$ and $p_t(\mathbf{x} \mid \mathbf{x}_1) := q_t(\mathbf{x} \mid \mathbf{x}_1)$. Then*

$$\mathcal{E}_t(\mathbf{x}) = -\frac{1}{k}\log \mathbb{E}_{q_t(\mathbf{x}_1 \mid \mathbf{x})}\left[e^{-k\mathcal{E}_1(\mathbf{x}_1)}\right] + const, \tag{10}$$

*and the guided distribution $p_t(\mathbf{x})$ will be generated by the flow*

$$\widehat{u}_t(\mathbf{x}) = \frac{\int_{\mathbf{x}_1} q_t(\mathbf{x}_1 \mid \mathbf{x}) \, u_t(\mathbf{x} \mid \mathbf{x}_1) \exp(-k\mathcal{E}_1(\mathbf{x}_1)) \, d\mathbf{x}_1}{\int_{\mathbf{x}_1} q_t(\mathbf{x}_1 \mid \mathbf{x}) \exp(-k\mathcal{E}_1(\mathbf{x}_1)) \, d\mathbf{x}_1}, \tag{11}$$

*which will generate final distribution $p_1(\mathbf{x}) \propto q_1(\mathbf{x}) \exp(-k\mathcal{E}_1(\mathbf{x}))$.*

Theorem 3.1 provide a method to construct the vector field $\widehat{u}_t(\mathbf{x})$ from the conditional vector field $u_t(\mathbf{x}|\mathbf{x}_1)$ and the intermediate energy function $\mathcal{E}_t$ in the closed-form solution. Since we have learned the vector field $v_t(\mathbf{x})$, we have the following corollary.

**Corollary 3.2** *Given an energy function $\mathcal{E}_1(\cdot)$ and a trained flow $v_t(\mathbf{x})$ that generates the probability distribution $q_t(\mathbf{x})$. Following the presumption in Theorem 3.1, then the guided distribution $p_t(\mathbf{x}) \propto q_t(\mathbf{x}) \exp(-k\mathcal{E}_t(\mathbf{x}))$ is generated by the flow*

$$\widehat{u}_t(\mathbf{x}) = v_t(\mathbf{x}) + r_t(\mathbf{x}), \tag{12}$$

*where $r_t(\mathbf{x}) = \frac{\int_{\mathbf{x}_1} q_t(\mathbf{x}_1|\mathbf{x})\zeta(\mathbf{x},\mathbf{x}_1)\exp(-k\mathcal{E}_1(\mathbf{x}_1))d\mathbf{x}_1}{\int_{\mathbf{x}_1} q_t(\mathbf{x}_1|\mathbf{x})\exp(-k\mathcal{E}_1(\mathbf{x}_1))d\mathbf{x}_1}$ and $\zeta(\mathbf{x},\mathbf{x}_1) = u_t(\mathbf{x} \mid \mathbf{x}_1) - v_t(\mathbf{x})$.*

As shown in Theorem 3.1 and Corollary 3.2, incorporating the energy term into the learned flow can be viewed as adding a corrective vector field $r_t(\mathbf{x})$ to the baseline vector field $v_t(\mathbf{x})$:

$$\widehat{u}_t(\mathbf{x}) = v_t(\mathbf{x}) + r_t(\mathbf{x}). \tag{13}$$

**Algorithm 1:** Training Procedure (Single Epoch) for Force Model

---

**Input:** trained flow network $v$, energy function $\mathcal{E}_1(\cdot)$, guided network $h_\theta$

**foreach** $x_1, y$ *in training set* **do**

    Randomly perturb $\mathbf{x}_1$ to obtain $\mathbf{x}_a^{1:K}$

    Sample $\mathbf{x}_0 = (C_0, O_0, \boldsymbol{\tau}_0) \sim p_0$, $t \sim \mathcal{U}(0,1)$;

    Sample $\mathbf{x}_t \sim p_t(\mathbf{x} \mid \mathbf{x}_0, \mathbf{x}_1) = p_t(C \mid C_0, C_1) \times p_t(O \mid O_0, O_1) \times p_t(\boldsymbol{\tau} \mid \boldsymbol{\tau}_0, \boldsymbol{\tau}_1)$;

    Set $w_i = \dfrac{\exp\left(\log q_t(\mathbf{x}_a^i \mid \mathbf{x}_t) - k\mathcal{E}_1(\mathbf{x}_a^i)\right)}{\sum_{j=1}^K \exp\left(\log q_t(\mathbf{x}_a^j \mid \mathbf{x}_t) - k\mathcal{E}_1(\mathbf{x}_a^j)\right)}$,

        $\zeta_{i,\text{tr}} = u_{t,\text{tr}}\left(C_t \mid C_a^i\right) - v_{t,\text{tr}}(\mathbf{x}_t)$,

        $\zeta_{i,\text{rot}} = u_{t,\text{rot}}\left(O_t \mid O_a^i\right) - v_{t,\text{rot}}(\mathbf{x}_t)$,

        $\zeta_{i,\text{tor}} = u_{t,\text{tor}}\left(\boldsymbol{\tau}_t \mid \boldsymbol{\tau}_a^i\right) - v_{t,\text{rot}}(\mathbf{x}_t)$;

    $L = \|h_{t,\text{tr}}(\mathbf{x}_t) - \sum_{i=1}^K w_i \zeta_{i,\text{tr}}\|_2^2 +$

        $\|h_{t,\text{rot}}(\mathbf{x}_t) - \sum_{i=1}^K w_i \zeta_{i,\text{rot}}\|_{SO(3)}^2 +$

        $\|h_{t,\text{tor}}(\mathbf{x}_t) - \sum_{i=1}^K w_i \zeta_{i,\text{tor}}\|_2^2$;

    Update $\theta \leftarrow \theta - \eta \nabla_\theta L$.

---

The correction term $r_t(\mathbf{x})$ accounts for how likely each conformation $\mathbf{x}_1$ is, weighted by $\exp[-k\,\mathcal{E}_1(\mathbf{x}_1)]$. In practice, we introduce an additional *force network* $h_t(\mathbf{x})$ to approximate $r_t(\mathbf{x})$, enabling efficient end-to-end training.

Algorithm 1 outlines the single-epoch training procedure. For each training complex $(\mathbf{x}_1, \mathbf{y})$, we sample $K$ candidate ligand conformations from the random perturbations of $\mathbf{x}_1$ and compute weighted residuals $\zeta_i$ based on $\exp[-k\,\mathcal{E}_1(\mathbf{x}_a^i)]$. The objective is to match $h_t(\mathbf{x}_t)$ to the weighted average of these residuals. Here, the perturbed data is sampled from $p_{1-\delta}(\mathbf{x} \mid \mathbf{x}_1)$, with $\delta \in \mathcal{U}(0, 0.1)$ and $K$ is a critical hyper-parameter in our method, as it directly affects the accuracy of the energy landscape estimation during training. A higher $K$ leads to a more accurate approximation of this distribution and, consequently, more precise force estimation.

After training, samples can be generated by integrating the modified ODE $\frac{d}{dt}\mathbf{x} = v_t(\mathbf{x}) + \eta h_t(\mathbf{x})$, where $\eta$ is a user-defined guidance strength. Larger $\eta$ emphasizes physical energy constraints more strongly, reducing the risk of steric clashes but potentially sacrificing some flexibility.

## 4 Experimental Results

### 4.1 Experiment Settings

**Dataset:** We utilized protein-ligand complexes from PDBBind, originating from the Protein Data Bank (PDB) [35]. Adopting the time-based splitting in [6], we trained our model on 17k complexes up to 2018 and tested on 363 structures in 2019, ensuring no ligand overlaps. This temporal split is favored over molecular scaffold or protein similarity-based methods [36, 15]. Further details about the pre-processing steps and construction of heterogeneous graphs are presented in Appendix B.

**Evaluating metrics:** We follow prior works [6, 10] and use ligand root-mean-square deviation (RMSD) of heavy atomic positions and centroid distance to compare predicted binding structures with ground-truth. The Ligand RMSD calculates the normalized Frobenius norm of the two corresponding matrices of ligand coordinates. The centroid distance is defined as the distance between the averaged 3D coordinates of the predicted and ground-truth bound ligand atoms. In addition, we use the PoseBuster [13] to validate chemical consistency and physical plausibility of the generated ligands.

**Implementation Details:** All statistics are averaged over three random seeds. We implemented all the models using the open-source Python library PyTorch and e3nn [37], and the experiments were conducted on a PC equipped with 4 NVIDIA A100-40GB GPUs. The network structures for flow matching model and guidance model are presented in Appendix C and the training and inference details with hyper-parameters are listed in Appendix D. In the inference stage, to generate an initial ligand conformation, we employed the ETKDG algorithm using RDKit, which randomly produces a low-energy ligand conformation. we follow common blind docking practice by first predicting

| Method | Ligand RMSD | | | | | | Centroid Distance | | | | | | Average |
|---|---|---|---|---|---|---|---|---|---|---|---|---|---|
| | Percentiles↓ | | | | % Below↑ | | Percentiles↓ | | | | % Below↑ | | |
| | 25% | 50% | 75% | Mean | 2Å | 5 Å | 25% | 50% | 75% | Mean | 2Å | 5Å | Runtime (s) |
| QVINA-W | 2.5 | 7.7 | 23.7 | 13.6 | 20.9 | 40.2 | 0.9 | 3.7 | 22.9 | 11.9 | 41.0 | 54.6 | 49* |
| GNINA | 2.8 | 8.7 | 22.1 | 13.3 | 21.2 | 37.1 | 1.0 | 4.5 | 21.2 | 11.5 | 36.0 | 52.0 | 146* |
| SMINA | 3.8 | 8.1 | 17.9 | 12.1 | 13.5 | 33.9 | 1.3 | 3.7 | 16.2 | 9.8 | 38.0 | 55.9 | 146 |
| GLIDE | 2.6 | 9.3 | 28.1 | 16.2 | 21.8 | 33.6 | 0.8 | 5.6 | 26.9 | 14.4 | 36.1 | 48.7 | 1405* |
| VINA | 5.7 | 10.7 | 21.4 | 14.7 | 5.5 | 21.2 | 1.9 | 6.2 | 20.1 | 12.1 | 26.5 | 47.1 | 205* |
| EQUIBIND | 3.8 | 6.2 | 10.3 | 8.2 | 5.5 | 39.1 | 1.3 | 2.6 | 7.4 | 5.6 | 40.0 | 67.5 | **0.03** |
| TANKBind | 2.4 | 4.0 | 7.7 | 7.4 | 19.3 | 61.7 | 0.9 | 1.7 | 4.2 | 5.5 | 56.5 | 77.4 | 0.87 |
| E3Bind | 2.1 | 3.8 | 7.8 | 7.2 | 23.4 | 60.0 | 0.8 | 1.5 | 4.0 | 5.1 | 60.0 | 78.8 | 0.44 |
| FABind | 1.7 | 3.1 | 6.7 | 6.4 | 33.1 | 64.2 | 0.7 | 1.3 | 3.6 | 4.7 | 60.3 | 80.2 | 0.12 |
| FABind+ | 1.3 | 2.4 | 5.3 | 5.1 | 43.8 | 73.3 | 0.5 | 1.0 | 2.6 | 3.5 | 69.1 | 86.2 | 6.4 |
| DiffDock | 1.4 | 3.6 | 8.0 | 7.5 | 38.4 | 62.4 | 0.5 | 1.3 | 3.2 | 5.5 | 60.8 | 79.0 | 68.1 |
| DiffDock + Vina | 1.3 | 3.2 | 5.5 | 3.8 | 43.5 | 73.0 | 0.3 | 1.2 | 3.0 | 4.1 | 66.1 | 88.3 | 45.6 |
| DiffDock + Conf | 1.3 | 3.3 | 5.5 | 4.3 | 42.6 | 69.1 | 0.4 | 1.1 | 3.1 | 4.5 | 65.8 | 83.2 | 45.5 |
| DiffDock + Gnina | 1.3 | 3.1 | 4.9 | 4.2 | 43.5 | 70.5 | 0.4 | 1.1 | 2.9 | 4.4 | 67.7 | 86.5 | 45.5 |
| DiffDock + Glide | 1.2 | 3.1 | 5.7 | 4.0 | 45.8 | 71.2 | 0.4 | 1.1 | 3.2 | 4.3 | 67.4 | 85.7 | 45.4 |
| Ours | 1.3 | 2.3 | 5.3 | 4.2 | 41.1 | 73.7 | 0.5 | 1.0 | 2.5 | 3.1 | 71.7 | 89.4 | 46.9 |
| Ours + Vina | **1.1** | 2.2 | 4.3 | 3.8 | 48.6 | 78.0 | 0.3 | 0.9 | 2.2 | 2.4 | 76.2 | **93.5** | 25.3 |
| Ours + Conf | 1.2 | 2.3 | 5.4 | 3.3 | 49.1 | 76.5 | 0.3 | **0.8** | 2.2 | 2.2 | 75.1 | 92.5 | 25.1 |
| Ours + Gnina | 1.2 | 2.2 | 5.2 | **3.1** | 47.5 | 75.6 | 0.4 | 0.9 | 2.0 | 2.4 | 74.3 | 91.6 | 25.1 |
| Ours + Glide | 1.2 | **2.1** | **4.2** | 3.2 | **49.5** | **78.0** | **0.3** | 0.9 | **2.1** | 1.9 | **77.1** | 92.5 | 25.2 |

Table 1: PDBBind blind Top-1 self-docking performance. The first quarter contains the results from traditional docking software, and the second contains recent deep learning-based docking methods . The last two lines show the force-guided results of DiffDock and our baseline model. The symbol "*" means that the method operates exclusively on the CPU. The best results are shown in **bold**.

binding pockets with P2Rank [38] and initializing ligand conformations around predicted centers. 40 poses are sampled in the inference stage and they are ranked by using the rank model in [10].

## 4.2 Molecular Docking Quality Assessment

In our assessment of molecular docking quality, we benchmarked our proposed method against several well-established methods, including SMINA [39], QuickVina-W [40], GLIDE [4], GNINA [14], Autodock Vina [3], EquiBind [6], TANKBind [15], E3Bind [7], DiffDock [10], FABind [8], FABind+ [12]. To demonstrate both the absolute performance of our model and the generality of the proposed force-guided sampling framework, we evaluate two families of methods (1) Our base model (Ours) and its energy-guided variants, and (2) DiffDock with exactly the same guidance procedure, detail for sampling with force guided diffusion model is presented in Appendix F. This section reports the main results for comparison, the additional results on the sensitive analysis on guidance strength $\eta$, number of generative samples are presented in appendix E.

### 4.2.1 Blind Self-Docking Performance for Seen and Unseen Proteins

Blind self-docking involves docking a flexible ligand to a protein without prior knowledge of the binding site, which requires accurate prediction of the ligand's conformation. We observe the notable performance of our ForceFM as shown in Table 1. By introducing force guidance strategies, both the DiffDock and our baseline models see notable improvements. Specifically, using force guidance results in a marked improvement in RMSD and centroid distance metrics. For example, one of the most notable results occurs with Ours + Glide, where the model achieves a mean RMSD of 3.2Å, a 49.5% success rate below 2Å, and an impressive 78.0% success rate under the 5Å threshold. These figures underscore the positive impact of integrating force guidance on docking accuracy. This model's performance is superior not only in RMSD but also in the overall prediction accuracy when compared to both the baseline models and traditional docking approaches such as Vina and DiffDock. When comparing results across models, it is evident that integrating force-guided strategies (either with Vina, Conf, Gnina, or Glide) improves both the percentile RMSD and the percent below 2Å and

| Method | Ligand RMSD | | | | | | Centroid Distance | | | | | |
| | Percentiles↓ | | | | % Below↑ | | Percentiles↓ | | | | % Below↑ | |
| | 25% | 50% | 75% | Mean | 2Å | 5 Å | 25% | 50% | 75% | Mean | 2Å | 5Å |
|---|---|---|---|---|---|---|---|---|---|---|---|---|
| QVINA-W | 3.4 | 10.3 | 28.1 | 16.9 | 15.3 | 31.9 | 1.3 | 6.5 | 26.8 | 15.2 | 35.4 | 47.9 |
| GNINA | 4.5 | 13.4 | 27.8 | 16.7 | 13.9 | 27.8 | 2.0 | 10.1 | 27.0 | 15.1 | 25.7 | 39.5 |
| SMINA | 4.8 | 10.9 | 26.0 | 15.7 | 9.0 | 25.7 | 1.6 | 6.5 | 25.7 | 13.6 | 29.9 | 41.7 |
| GLIDE | 3.4 | 18.0 | 31.4 | 19.6 | 19.6 | 28.7 | 1.1 | 17.6 | 29.1 | 18.1 | 29.4 | 40.6 |
| VINA | 7.9 | 16.6 | 27.1 | 18.7 | 1.4 | 12.0 | 2.4 | 15.7 | 26.2 | 16.1 | 20.4 | 37.3 |
| EQUIBIND | 5.9 | 9.1 | 14.3 | 11.3 | 0.7 | 18.8 | 2.6 | 6.3 | 12.9 | 8.9 | 16.7 | 43.8 |
| TANKBind | 3.4 | 5.7 | 10.8 | 10.5 | 3.5 | 43.7 | 1.2 | 2.6 | 8.4 | 8.2 | 40.9 | 70.8 |
| E3Bind | 3.0 | 6.1 | 10.2 | 10.1 | 6.3 | 38.9 | 1.2 | 2.3 | 7.0 | 7.6 | 43.8 | 66.0 |
| FABind | 2.2 | 3.4 | 8.3 | 7.7 | 19.4 | 60.4 | 0.9 | 1.5 | 4.7 | 5.9 | 57.6 | 75.7 |
| FABind+ | 1.7 | 2.9 | 8.4 | 7.2 | 33.3 | 63.9 | 0.8 | 1.5 | 4.6 | 5.4 | 59.7 | 77.1 |
| DiffDock | 2.8 | 6.4 | 16.3 | 12.0 | 17.2 | 42.3 | 1.0 | 2.7 | 14.2 | 9.8 | 43.3 | 62.6 |
| DiffDock + Vina | 2.6 | 5.8 | 12.5 | 9.7 | 23.8 | 49.9 | 0.9 | 2.5 | 12.1 | 8.9 | 47.6 | 70.2 |
| DiffDock + Conf | 2.7 | 5.9 | 13.2 | 10.1 | 20.1 | 46.4 | 1.0 | 2.6 | 12.4 | 9.6 | 48.4 | 70.5 |
| DiffDock + Gnina | 2.4 | 5.4 | 11.8 | 9.3 | 25.1 | 50.2 | 0.9 | 2.5 | 11.3 | 9.0 | 50.5 | 72.3 |
| DiffDock + Glide | 2.5 | 5.6 | 12.0 | 9.5 | 25.4 | 49.1 | 1.0 | 2.4 | 11.6 | 8.9 | 49.2 | 71.0 |
| Ours | 1.7 | 3.2 | 7.3 | 5.7 | 31.8 | 62.8 | 0.7 | 1.4 | 4.0 | 4.6 | 63.2 | 78.3 |
| Ours + Vina | 1.5 | 2.7 | **6.5** | 4.9 | 38.6 | 69.5 | 0.5 | 1.3 | 3.5 | 3.2 | 70.2 | 84.9 |
| Ours + Conf | 1.7 | 3.2 | 7.2 | 5.6 | 32.7 | 63.7 | 0.7 | 1.4 | 3.9 | 4.4 | 65.2 | 79.5 |
| Ours + Gnina | 1.5 | 2.8 | **6.5** | 4.8 | **40.1** | 69.5 | 0.6 | **1.2** | **3.2** | 3.3 | 70.2 | 83.4 |
| Ours + Glide | **1.4** | **2.7** | 6.6 | **4.8** | 39.7 | **70.3** | **0.5** | 1.3 | 3.5 | **3.1** | **71.7** | **85.2** |

Table 2: Performance of blind self-docking on unseen receptors.

5Å. The Ours + Glide method, for instance, produces some of the best performance, especially with a remarkable 77.1% success rate for predictions within 5Å and a 92.5% success rate for predictions within 5Å when including Conf guidance.

In conclusion, incorporating force guidance significantly enhances the performance of self-docking, offering more reliable and accurate predictions for ligand-protein interactions, especially when there is no prior knowledge of the binding site. This approach represents a key advancement in the field of computational docking and ligand-protein interaction prediction.

### 4.2.2 Blind Self-Docking Performance for Unseen Proteins

This section evaluates the generalization capability of our model on proteins not seen during training. Following the approach of previous studies [8, 12], we assess the model's performance on a set of proteins filtered by their UniProt IDs, excluding those encountered during training or validation. The evaluation results, summarized in Table 2, provide an in-depth analysis of the model's performance on these unseen proteins.

Based on the data in Table 2, the force-guided models significantly outperform other methods on unseen proteins. Specifically, the Ours + Glide model achieves the best performance with a 4.8Å mean RMSD and success rates of 39.7% at 2Å and 70.3% at 5Å, surpassing all baseline methods, especially in terms of precision.

### 4.2.3 PoseBuster Evaluation

In addition to conventional metrics, we employed the PoseBuster benchmark (Fig.2(a)) to assess the physical validity of the generated ligand poses, which is crucial for their practical applicability. While standard metrics such as RMSD provide spatial alignment evaluations, the PoseBuster benchmark further examines the generated poses for physical and chemical plausibility. It imposes specific criteria, such as avoiding steric clashes and maintaining realistic bond lengths and angles, to ensure that predicted poses are biologically viable.

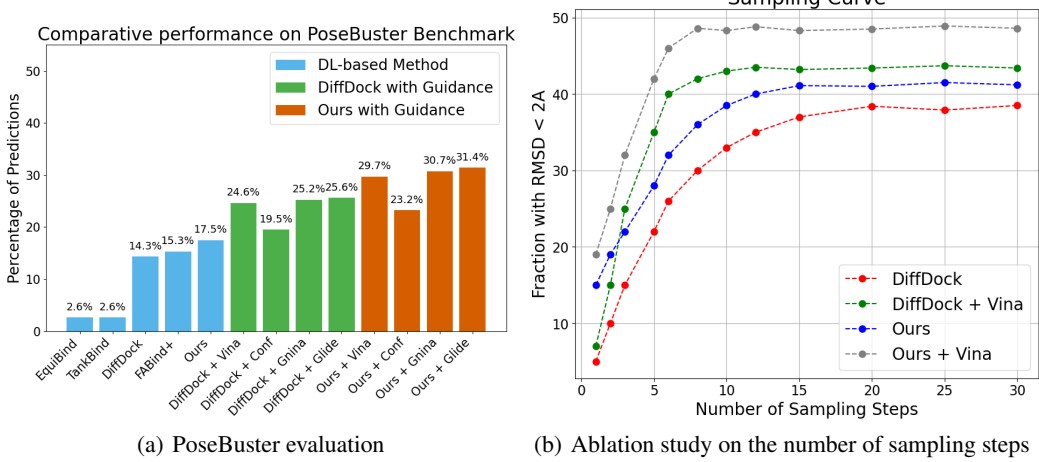

(a) PoseBuster evaluation

(b) Ablation study on the number of sampling steps

Figure 2: (a) PoseBuster evaluation: The model with force guidance significantly outperform the model without guidance. (b) Ablation study on the number of sampling steps.

Figure 2(a) reveals a clear stratification among the evaluated pipelines. The pure deep-learning (DL) baselines methods including "Ours" show poor energetic reasoning, underscoring the pitfalls of relying on geometric alignment alone. Crucially, supplementing both DiffDock and our method with *force guidance* produces a dramatic jump in physical validity. For DiffDock, the best guidance option (*+Glide*) lifts the pass rate from 14.3% to 25.6%—an absolute gain of 11.3% and a relative improvement of about 79%. Our method benefits even more: the same Glide-based refinement boosts performance from 17.5% to 31.4% (+13.9%, also 79% relative). Across all four guidance strategies (Vina, ConfGen, Gnina, Glide), our approach consistently stays 4–6% ahead of the corresponding DiffDock variant, establishing a new state-of-the-art with up to 31.4% PoseBuster-compliant predictions. These findings demonstrate that the energetic guidance is indispensable for converting generated poses into physically realistic, drug-like conformations.

### 4.2.4 Performance with Number of Sampling Steps.

Figure 2(b) illustrates the relationship between the number of sampling steps and the fraction of docking results with RMSD $\leq 2$Åacross four methods: DiffDock, DiffDock + Vina, Ours, and Ours + Vina. The key observation is that energy-based guidance significantly improves the convergence speed and performance of generative sampling. DiffDock alone converges around 20 samples, reaching a plateau in the success rate beyond this point. DiffDock + Vina, which incorporates energy-based sampling, achieves similar performance with 5 steps, converging around 12 steps. The method Ours without force guidance converges slightly faster than DiffDock, around 14 steps. Ours + Vina, combining our generative approach with Vina scoring, shows the fastest convergence, reaching a performance plateau at approximately 8.

These results demonstrate that integrating energy-based guidance significantly enhances sampling efficiency, reducing the number of required steps to achieve optimal docking performance.

## 5   Conclusion

In this work, we introduce ForceFM, a novel force-guided flow-matching model for protein-ligand docking. By integrating generative model with physical principles, ForceFM addresses key limitations of current state-of-the-art (SOTA) methods, particularly their struggles with physically consistency and structural validity. The incorporation of a force-guided network enables ForceFM to significantly outperform existing models in both accuracy and physical plausibility with fewer sampling steps. Extensive experiments on the PDBBind dataset and PoseBuster benchmark demonstrate its superior performance, yielding physically consistent and structurally valid ligand conformations. Additionally,

ForceFM shows strong generalization to unseen protein data, making it a versatile and reliable tool for drug discovery and molecular docking.

However, there are still certain limitations. Despite its promising results, ForceFM relies on the energy functions chosen for force guidance, and performance may vary depending on the choice of energy function. Furthermore, while the model successfully handles static protein-ligand interactions, it does not yet fully account for protein flexibility or the dynamic conformational changes that occur during ligand binding. In addition, while the force-guided network improves physical plausibility and data efficiency, its generalization ability remains constrained by the training data. ForceFM is primarily trained on the PDBBind dataset, which limits its exposure to diverse protein families, binding environments, and rare conformational states. This can affect performance when applied to novel targets or large-scale virtual screening tasks.

In future work, we plan to enhance ForceFM by incorporating molecular dynamics simulations to capture protein flexibility and the conformational transitions in the protein-ligand complex during binding, and expand training across larger and more diverse datasets. These directions will enhance the model's robustness, improve generalization, and allow a more realistic description of protein-ligand interactions.

## Acknowledgments and Disclosure of Funding

The authors acknowledge the financial supported by the Natural Science Foundation of China (Grant No. 12371290).

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

# Appendix

## A  Proof of Theorem 3.1

Before proof, we need the following lemma:

**Lemma A.1 (Theorem 3.1 in [27])** *Let $q_t(\mathbf{x}_t)$ be the data-based marginal distribution on specific protein* **y**. *Suppose $p_t(\mathbf{x}_t \mid \mathbf{x}_1) := q_t(\mathbf{x}_t \mid \mathbf{x}_1)$ and $p_t(\mathbf{x}_t) \propto q_t(\mathbf{x}_t) e^{-k\mathcal{E}_t(\mathbf{x}_t)}$, for $t \in [0,1]$. Then $\mathcal{E}_t(\mathbf{x}_t)$ satisfies*

$$\mathcal{E}_t(\mathbf{x}_t) = -\frac{1}{k} \log \mathbb{E}_{q_t(\mathbf{x}_1 \mid \mathbf{x}_t)}\left[e^{-k\mathcal{E}_1(\mathbf{x}_1)}\right], \tag{14}$$

*after dropping the constant, and*

$$\nabla_{\mathbf{x}_t}\mathcal{E}_t(\mathbf{x}_t) = \frac{\mathbb{E}_{q_1(\mathbf{x}_1)}\left[q_t(\mathbf{x}_t \mid \mathbf{x}_1) e^{-k\mathcal{E}_1(\mathbf{x}_1)} G(\mathbf{x}_1, \mathbf{x}_t)\right]}{k\mathbb{E}_{q_1(\mathbf{x}_1)}\left[q_t(\mathbf{x}_t \mid \mathbf{x}_1) e^{-k\mathcal{E}_1(\mathbf{x}_1)}\right]}, \tag{15}$$

*with $G(\mathbf{x}_1, \mathbf{x}_t) := \nabla_{\mathbf{x}_t} \log q_t(\mathbf{x}_t) - \nabla_{\mathbf{x}_t} \log q_t(\mathbf{x}_t \mid \mathbf{x}_1)$.*

**Proof of Lemma A.1** Given $p_1(\mathbf{x}_1) = q_1(\mathbf{x}_1) e^{-k\mathcal{E}_1(\mathbf{x}_1)}/Z$ and $p_t(\mathbf{x}_t \mid \mathbf{x}_1) = q_t(\mathbf{x}_t \mid \mathbf{x}_1)$, the distribution $p_t(\mathbf{x}_t)$ follows

$$
\begin{aligned}
p_t(\mathbf{x}_t) &= \int p_t(\mathbf{x}_t \mid \mathbf{x}_1) p_1(\mathbf{x}_1)\, \mathrm{d}\mathbf{x}_1 = \int q_t(\mathbf{x}_t \mid \mathbf{x}_1) \frac{q_1(\mathbf{x}_1) e^{-k\mathcal{E}_1(\mathbf{x}_1)}}{Z}\, \mathrm{d}\mathbf{x}_1 \\
&= \int q_t(\mathbf{x}_1 \mid \mathbf{x}_t) q_t(\mathbf{x}_t) \frac{e^{-k\mathcal{E}_1(\mathbf{x}_1)}}{Z}\, \mathrm{d}\mathbf{x}_1 = q_t(\mathbf{x}_t) \mathbb{E}_{q_t(\mathbf{x}_1 \mid \mathbf{x}_t)}\left[\frac{e^{-k\mathcal{E}_1(\mathbf{x}_1)}}{Z}\right].
\end{aligned}
\tag{16}
$$

Under the assumption that

$$p_t(\mathbf{x}_t) := q_t(\mathbf{x}_t) e^{-k\mathcal{E}_t(\mathbf{x}_t)}, \tag{17}$$

then $\mathcal{E}_t(\mathbf{x}_t)$ satisfies

$$\mathcal{E}_t(\mathbf{x}_t) = -\frac{1}{k} \log \mathbb{E}_{q_t(\mathbf{x}_1 \mid \mathbf{x}_t)}\left[e^{-k\mathcal{E}_1(\mathbf{x}_1)}\right] + \frac{1}{k} \log Z, \tag{18}$$

for $0 < t \leq 1$.

The corresponding intermediate force can be derived by

$$\nabla_{\mathbf{x}_t}\mathcal{E}_t(\mathbf{x}_t) = -\frac{\int e^{-k\mathcal{E}_1(\mathbf{x}_1)} \nabla_{\mathbf{x}_t} q_t(\mathbf{x}_1 \mid \mathbf{x}_t)\, \mathrm{d}\mathbf{x}_1}{k \int q_t(\mathbf{x}_1 \mid \mathbf{x}_t) e^{-k\mathcal{E}_1(\mathbf{x}_1)} \mathrm{d}\mathbf{x}_1}. \tag{19}$$

The numerator can be derived as

$$
\begin{aligned}
&\int e^{-k\mathcal{E}_1(\mathbf{x}_1)} \nabla_{\mathbf{x}_t} q_t(\mathbf{x}_1 \mid \mathbf{x}_t)\, \mathrm{d}\mathbf{x}_1 \\
&= \int e^{-k\mathcal{E}_1(\mathbf{x}_1)} q_t(\mathbf{x}_1 \mid \mathbf{x}_t) \nabla_{\mathbf{x}_t} \log q_t(\mathbf{x}_1 \mid \mathbf{x}_t)\, \mathrm{d}\mathbf{x}_1 \\
&= \int q_t(\mathbf{x}_1 \mid \mathbf{x}_t) e^{-k\mathcal{E}_1(\mathbf{x}_1)} \nabla_{\mathbf{x}_t} \log \frac{q_t(\mathbf{x}_t \mid \mathbf{x}_1) q_1(\mathbf{x}_1)}{q_t(\mathbf{x}_t)} \mathrm{d}\mathbf{x}_1 \\
&= \int q_t(\mathbf{x}_1 \mid \mathbf{x}_t) e^{-k\mathcal{E}_1(\mathbf{x}_1)} \left(\nabla_{\mathbf{x}_t} \log q_t(\mathbf{x}_t \mid \mathbf{x}_1) - \nabla_{\mathbf{x}_t} \log q_t(\mathbf{x}_t)\right) \mathrm{d}\mathbf{x}_1 \\
&= -\mathbb{E}_{q_t(\mathbf{x}_1 \mid \mathbf{x}_t)}\left[e^{-k\mathcal{E}_1(\mathbf{x}_1)} \zeta(\mathbf{x}_1, \mathbf{x}_t)\right],
\end{aligned}
\tag{20}
$$

where $\zeta(\mathbf{x}_1, \mathbf{x}_t) := \nabla_{\mathbf{x}_t} \log q_t(\mathbf{x}_t) - \nabla_{\mathbf{x}_t} \log q_t(\mathbf{x}_t \mid \mathbf{x}_1)$. Therefore

$$
\begin{aligned}
\nabla_{\mathbf{x}_t}\mathcal{E}_t(\mathbf{x}_t) &= \frac{\mathbb{E}_{q_t(\mathbf{x}_1 \mid \mathbf{x}_t)}\left[e^{-k\mathcal{E}_1(\mathbf{x}_1)} \zeta(\mathbf{x}_1, \mathbf{x}_t)\right]}{k\mathbb{E}_{q_t(\mathbf{x}_1 \mid \mathbf{x}_t)}\left[e^{-k\mathcal{E}_1(\mathbf{x}_1)}\right]} \\
&= \frac{\mathbb{E}_{q_1(\mathbf{x}_1)}\left[q_t(\mathbf{x}_t \mid \mathbf{x}_1) e^{-k\mathcal{E}_1(\mathbf{x}_1)} \zeta(\mathbf{x}_1, \mathbf{x}_t)\right]}{k\mathbb{E}_{q_1(\mathbf{x}_1)}\left[q_t(\mathbf{x}_t \mid \mathbf{x}_1) e^{-k\mathcal{E}_1(\mathbf{x}_1)}\right]}.
\end{aligned}
\tag{21}
$$

*Proof of Theorem 3.1.* Considering the dynamic of probability distribution $p_t(\mathbf{x}) = \frac{q_t(\mathbf{x})\mathcal{E}_t(\mathbf{x})}{Z}$, according to the continuity equation, we have

$$
\begin{aligned}
\frac{\mathrm{d}p_t(\mathbf{x})}{\mathrm{d}t} &= \frac{\mathrm{d}}{\mathrm{d}t} \frac{q_t(\mathbf{x})\exp\left(-k\mathcal{E}_t(\mathbf{x})\right)}{Z} \\
&= \frac{\mathrm{d}}{\mathrm{d}t} \frac{q_t(\mathbf{x})\int_{\mathbf{x}_1} q\left(\mathbf{x}_1 \mid \mathbf{x}\right)\exp\left(-k\mathcal{E}\left(\mathbf{x}_1\right)\right)\mathrm{d}\mathbf{x}_1}{Z} \\
&= \frac{\mathrm{d}}{\mathrm{d}t} \frac{\int_{\mathbf{x}_1} q_t\left(\mathbf{x}_1,\mathbf{x}\right)\exp\left(-k\mathcal{E}\left(\mathbf{x}_1\right)\right)\mathrm{d}\mathbf{x}_1}{Z} \\
&= \frac{\mathrm{d}}{\mathrm{d}t} \frac{\int_{\mathbf{x}_1} q_t\left(\mathbf{x} \mid \mathbf{x}_1\right) q\left(\mathbf{x}_1\right)\exp\left(-k\mathcal{E}\left(\mathbf{x}_1\right)\right)\mathrm{d}\mathbf{x}_1}{Z} \\
&= \frac{1}{Z} \int_{\mathbf{x}_1} \frac{\mathrm{d}}{\mathrm{d}t} q_t\left(\mathbf{x} \mid \mathbf{x}_1\right) q\left(\mathbf{x}_1\right)\exp\left(-k\mathcal{E}\left(\mathbf{x}_1\right)\right)\mathrm{d}\mathbf{x}_1.
\end{aligned}
\tag{22}
$$

In addition,

$$
\frac{\mathrm{d}q_t\left(\mathbf{x} \mid \mathbf{x}_1\right)}{\mathrm{d}t} = -\operatorname{div} \cdot \left[q_t\left(\mathbf{x} \mid \mathbf{x}_1\right) \mathbf{u}_t\left(\mathbf{x} \mid \mathbf{x}_1\right)\right].
\tag{23}
$$

Plugging the definition of continuity equation 23 into 22 we can obtain that

$$
\begin{aligned}
\frac{\mathrm{d}p_t(\mathbf{x})}{\mathrm{d}t} &= \frac{1}{Z} \int_{\mathbf{x}_1} \frac{\mathrm{d}}{\mathrm{d}t} q_t\left(\mathbf{x} \mid \mathbf{x}_1\right) q\left(\mathbf{x}_1\right)\exp\left(-k\mathcal{E}\left(\mathbf{x}_1\right)\right)\mathrm{d}\mathbf{x}_1 \\
&= -\frac{\int_{\mathbf{x}_1} \operatorname{div} \cdot \left[q_t\left(\mathbf{x} \mid \mathbf{x}_1\right) \mathbf{u}_t\left(\mathbf{x} \mid \mathbf{x}_1\right)\right] q\left(\mathbf{x}_1\right)\exp\left(-k\mathcal{E}\left(\mathbf{x}_1\right)\right)\mathrm{d}\mathbf{x}_1}{Z} \\
&= -\frac{\operatorname{div} \cdot \left[\int_{\mathbf{x}_1} q_t\left(\mathbf{x} \mid \mathbf{x}_1\right) \mathbf{u}_t\left(\mathbf{x} \mid \mathbf{x}_1\right) q\left(\mathbf{x}_1\right)\exp\left(-k\mathcal{E}\left(\mathbf{x}_1\right)\right)\mathrm{d}\mathbf{x}_1\right]}{Z} \\
&= -\operatorname{div} \cdot \left[\frac{q_t(\mathbf{x})\exp\left(-k\mathcal{E}_t(\mathbf{x})\right)}{Z} \int_{\mathbf{x}_1} q_t\left(\mathbf{x}_1 \mid \mathbf{x}\right) \mathbf{u}_t\left(\mathbf{x} \mid \mathbf{x}_1\right) \frac{\exp\left(-k\mathcal{E}\left(\mathbf{x}_1\right)\right)}{\exp\left(-k\mathcal{E}_t(\mathbf{x})\right)}\mathrm{d}\mathbf{x}_1\right] \\
&= -\operatorname{div} \cdot \left[p_t(\mathbf{x})\widehat{\mathbf{u}}_t(\mathbf{x})\right]
\end{aligned}
\tag{24}
$$

Thus we conclude our proof by showing $\widehat{\mathbf{u}}_t(\mathbf{x})$ defined by Eqn. (11) can generate the guided distribution sequence $p_t(\mathbf{x})$.

# B  Details for Constructing Protein, Ligand Features, and Heterogeneous Graphs

## B.1  Protein Representation

For protein representation, we utilize residue levels features. Each amino acid, denoted as $n_i^p$, is characterized by a feature vector $\mathbf{h}_i^p$, encoded using ESM2 [41] (*esm2_t33_650M_UR50D*), including its residue type. The position of each node, $\mathbf{x}_i^p$, corresponds to the Cartesian coordinates of the $C\alpha_i$ atom in $\mathbb{R}^3$. The total number of protein nodes is represented as $N^p$.

## B.2  Ligand Representation

The ligand is represented at the atom level, where each atom is considered a node with features such as atomic number, chirality, degree, formal charge, implicit valence, number of connected hydrogens, number of radical electrons, hybridization type, aromaticity, ring membership, and six ring-size indicators. Each atom's spatial coordinates are denoted as $x_j^l \in \mathbb{R}^3$. The total number of ligand nodes is given as $N^l$.

## B.3  Geometric heterogeneous graph construction

In this work, structures are represented as heterogeneous geometric graphs with nodes representing ligand (heavy) atoms, receptor residues (located in the position of the $\alpha$-carbon atom), receptor heavy

atoms. To build the radius graph, we connect nodes using cutoffs that are dependent on the types of nodes they are connecting:

- **Ligand atoms-Ligand atoms**: An edge between two ligand heavy atoms exists if the distance between them is less than 5Å or a covalent bond exists. We also represent covalent bonds with some initial embedding representing the bond type (single, double, triple, and aromatic).

- **Receptor residues-Receptor residues**: An edge between residual $v_k^p$ and $v_i^p$ exists if $v_k^p$ is among the 24 nearest neighbors of $v_i^p$ within 15Å.

- **Receptor residues-Ligand atoms**: It uses a cutoff of $20 + 3 * \sigma_{\text{tr}}$Å where $\sigma_{\text{tr}} = (1 - t)\sigma_{\text{tr, max}} + t\sigma_{\text{tr, min}}$ is the current standard deviation for translation.

All these features (protein, ligand nodes and edges) are concatenated with sinusoidal embeddings of the time. As for the edges, they are concatenated with radial basis embeddings of edge length. These scalar features of each node and edge are then transformed with learnable two-layer MLPs (different for each node and edge type) into a set of scalar features with number $ns$ that are used as initial representations by the following layers.

## C   Network Structures

In this part, we provide a detailed description of the network structures of each stage. We first introduce the core feature extraction module: E3-equivariant graph convolution layer (EGCL).

**EGCL: E3-equivariant graph convolution layer**

EGCL serves as the foundational module for feature extraction in our model, which is build upon Tensor field and e3nn [37]. It operates by utilizing the tensor products of current node features with the spherical harmonic representations of edge vectors to construct messages. These tensor products are weighted according to the edge embeddings and scalar features of the connected nodes. For any node $a$ in a set of nodes $\mathcal{G}_\mathcal{A}$, a radius graph is constructed with target nodes from $\mathcal{G}_\mathcal{B}$. The output feature for node $a$ is then formulated as follows:

$$out_a = \text{BN}\left(\frac{1}{|\mathcal{N}_a|}\sum_{b\in\mathcal{N}_a} Y\left(\hat{r}_{ab}\right)\otimes_{\psi_{ab}} \mathbf{h}_b\right) \tag{25}$$

with $\psi_{ab} = \Psi\left(e_{ab}, \mathbf{h}_a, \mathbf{h}_b\right)$, where $\mathcal{N}_a$ denotes the neighbors of $a$ in $\mathcal{G}_\mathcal{B}$, $\mathbf{h}_a$ and $\mathbf{h}_b$ are feature vectors, $e_{ab}$ is the edge embeddings of $(a, b)$, and $Y\left(\hat{r}_{ab}\right)$ represents the spherical harmonics of the edge direction vector up to $\ell = 2$. The BN symbolizes the equivariant batch normalization. The orders of the output are restricted to a maximum of $\ell = 1$. In addition, $\otimes_{\psi_{ab}}$ refers to the spherical tensor product of irreps with path weights $\psi_{ab}$, and all learnable weights are contained in $\Psi$, a dictionary of MLPs with dropout. This process generates scalar and vector representations for each node, leading to the output feature matrix as described in the following equation:

$$EGCL(\mathcal{G}_\mathcal{A}, \mathcal{G}_\mathcal{B}) = [out_1, out_2, \dots, out_{N_A}], \tag{26}$$

where $N_A$ is the number of nodes in $\mathcal{G}_\mathcal{A}$.

### C.1   Baseline Model

Bt stacking EGCLs, at time $t$, the current perturbed ligand graph is represent as $\mathcal{G}_{lig} = \{H^{lig}, X^{lig}\}$ and protein residue graph $\mathcal{G}_p = \{H^p, X^p\}$. The feature matrices are updated by using the EGCLs, which are shown as follows:

$$\begin{aligned} H_l^p =& f_{p,p}^l(\mathcal{G}_p^{l-1}, \mathcal{G}_p^{l-1}) + f_{p,lig}^l(\mathcal{G}_p^{l-1}, \mathcal{G}_{lig}^{l-1}) \\ &+ H_{l-1}^p, \end{aligned} \tag{27}$$

$$\begin{aligned} H_l^{lig} =& f_{lig,lig}^l(\mathcal{G}_{lig}^{l-1}, \mathcal{G}_{lig}^{l-1}) + f_{lig,p}^l(\mathcal{G}_{lig}^{l-1}, \mathcal{G}_p^{l-1}) \\ &+ H_{l-1}^{lig}, \end{aligned} \tag{28}$$

where the functions $f_{*,*}^l, l = 1, ..., L$ here are EGCLs. Finally, after obtaining high-quality features for ligand, these features are utilized to predict the translation, rotation, and torsion vector at current time $t$. Translation and rotation can intuitively represent the linear and angular accelerations of the ligand's center of mass and the remaining molecular structure, respectively.

**Translation and Rotation:** The translation and rotation of the ligand are represented as the linear and angular accelerations, respectively, of its molecular structure. Specifically, we aim to produce two output vectors, one each for translation and rotation. These vectors are generated by convolution each ligand atom's features with the ligand's center of mass, $c$. The process is mathematically expressed as:

$$\mathbf{v} \leftarrow \frac{1}{|\mathcal{V}_\ell|} \sum_{a \in \mathcal{V}_\ell} Y\left(\hat{r}_{ca}\right) \otimes_{\psi_{ca}} \mathbf{h}_a$$

$$\text{with } \psi_{ca} = \Psi\left(\mu\left(r_{ca}\right), \mathbf{h}_a\right)$$

(29)

where $\mu(\cdot)$ indicates radial basis embeddings of the edge length. The output vector $\mathbf{v}$ contains 2 odd and 2 even vectors (1 single odd and 1 even for translation and another 2 for rotation and they are summed). These vectors' magnitudes are fine-tuned using an MLP, which considers their current magnitudes and sinusoidal embeddings of the time. The final translation and rotation vectors are adjusted by multiplying by $1/\sigma_{tr}$ and $\sigma_{rot}$, respectively.

**Torsion:** The torsional aspect of ligand movement is defined based on its rotatable bonds. Assuming there are $m$ rotatable bonds in a ligand, we employ a pseudo-torque layer, akin to the approach in Jing et al. [34], to predict SE(3)-invariant scalars for each rotatable bond.

For a rotatable bond $g = (g_0, g_1)$ ($g_0$ and $g_1$ are two nodes on the bond) and node $b$ in the ligand graph $\mathcal{G}_{lig}$, the vectorial components $r_{gb}$ and $\hat{r}_{gb}$ denote the magnitude and direction connecting bond $g$'s center and atom $b$ respectively. A convolutional filter $T_g$, specific to each bond $g$, is constructed using:

$$T_g(\hat{r}) := Y^2\left(\hat{r}_g\right) \otimes Y(\hat{r})$$

(30)

where $\otimes$ represents the complete tensor product as detailed in [37], and the second term contains the spherical harmonics up to $\ell = 1$. This filter is then used to convolve with the representations of every neighbor on a radius graph:

$$\mathcal{E}_\tau = \{(g, b) \mid g \text{ a rotatable bond}, b \in \mathcal{V}_\ell\}$$

$$e_{gb} = F\left(\mu\left(r_{gb}\right)\right) \quad \forall (g, b) \in \mathcal{E}_\tau$$

$$\mathbf{h}_g = \frac{1}{|\mathcal{N}_g|} \sum_{b \in \mathcal{N}_g} T_g\left(\hat{r}_{gb}\right) \otimes_{\gamma_{gb}} \mathbf{h}_b$$

$$\text{with } \gamma_{gb} = G\left(e_{gb}, \mathbf{h}_b, \mathbf{h}_{g_0} + \mathbf{h}_{g_1}\right)$$

(31)

Here, $\mathcal{N}_g = \{b \mid (g, b) \in \mathcal{E}_\tau\}$, and $F$ and $G$ are MLPs with learnable parameters. Finally, we produce a single scalar prediction for each bond by using both odd and even representations:

$$out_g = F_{out}\left(\mathbf{h}_{g,odd}, \mathbf{h}_{g,even}\right)$$

(32)

where $F_{out}$ is a two-layer MLP with $\tanh$ non-linearity and no biases. This is also multiplied by $\sigma_{tor}$.

### C.2 Force Model

The force model has the same structure with baseline model, with a smaller network size.

## D Training & Inference Details with Hyper-parameters

**General Settings:** Both baseline model and force model employ the AdamW optimizer with a learning rate of $5e - 5$ and weight decay $1e - 4$. The learning rate is controlled by cosine annealing scheduler with minimum lr=$1e - 6$. During inference, we use the exponential moving average of weights, updated after each optimization step with a decay factor of 0.999.

For manifold flow matching, we actually randomly perturb $\mathbf{x}_1$ to get $\mathbf{x}_0$. The rotation vector can be obtained in the axis-angle parameterization by sampling a unit vector uniformly and random angle

omega in $[0, \pi]$ according to the following distribution.

$$p(\omega) = \frac{1 - \cos(\omega)}{\pi} f(\omega), \tag{33}$$

where $f(\omega) = \sum_{l=0}^{\infty} (2l+1) \exp(-l(l+1)\sigma_{rot,max}^2) \sin(\omega(l+1/2))/\sin(\omega/2)$. The translation perturbation kernel is normal with variance $\sigma_{tr,max}^2$ and torsion is warped normal with variance $\sigma_{tor,max}^2$. $\sigma_{tr,max} = 8.0, \sigma_{tr,min} = 0.1, \sigma_{rot,max} = \pi/2, \sigma_{rot,min} = \pi/100, \sigma_{tor,max} = \pi, \sigma_{tor,min} = \pi/100$.

### D.1 Details for Baseline Flow Matching Model

**Objective:** The baseline model aims for producing primitive estimation of ligand position.

**Training Protocol:**

- *Training Duration:* The model is trained over 2000 epochs on the PDBBind dataset, with a batch size of 8 and a dropout rate of 0.1.
- *Model Architecture:* The number of edge length embeddings is set to 64. We employ 5 EGCLs ($L = 6$), with 60 scalar features ($ns$) and 12 vector features ($nv$).
- *Inference Process:* Pockets are identified based on P2Rank [38]. By initializing ligand conformations around the pocket, we can sampling ligands via reverse diffusion. The reverse process is split into 15 time steps.

### D.2 Details for Force Model

**Training Settings:**

- *Extended Training:* The model involves 400 epochs of training, a batch size of 8, and a dropout rate of 0.1.
- *Model Complexity:* The model includes 4 EGCLs ($L = 4$), with each EGCL having 48 scalar ($ns$) and 10 vector ($nv$) features.
- *Hyper-parameters:* The inverse of the temperature $k = \frac{1}{10}, \eta = 1$. The number of samples $K$ for expectation approximation in Eqn.(12) is 40.

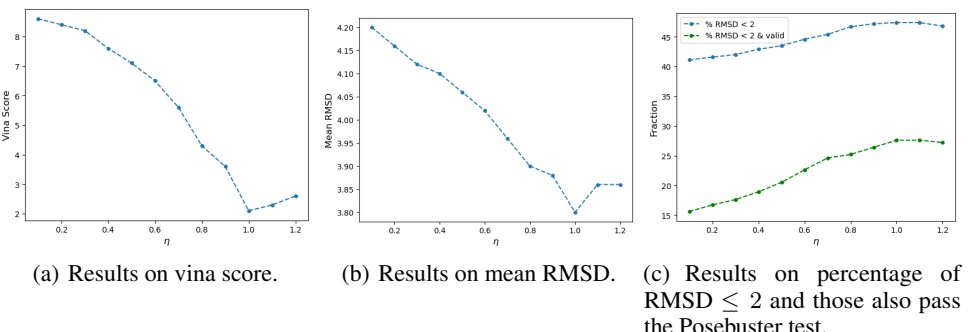

(a) Results on vina score.    (b) Results on mean RMSD.    (c) Results on percentage of RMSD $\leq 2$ and those also pass the Posebuster test.

Figure 3: Ablation study on $\eta$.

**Energy function/software details:** We run GLIDE, GNINA, and Autodock Vina with their default settings

- Autodock Vina is a well-established tool. Vina score is computed by following `https://autodock-vina.readthedocs.io/en/latest/docking_python.html`.
- Gnina builds on SMINA by additionally using a learned 3D CNN for scoring. We uses Gnina v1.3.2 in this work `https://github.com/gnina/gnina/releases/tag/v1.3.2`. We used the Vina-style scoring function implemented in gnina as score function in table 1 in main text. We conduct additional experiments using GNINA's CNN-based scoring functions (e.g., CNNscore, CNNaffinity) in table 7.

- ConfScore [10] is computed using version 1.1 `https://github.com/gcorso/DiffDock/releases/download/v1.1/diffdock_models.zip`.

- Glide is a strong heavily used commercial docking tool.

- P2Rank v2.3 is used for pocket identification `https://github.com/rdk/p2rank/releases/download/2.3/p2rank_2.3.tar.gz`.

# E  Additional Experimental Results

In this part, addition ablation studies are conducted for hyper parameter tuning. All these experiments are conducted by using force-guided model with energy function Vina score.

## E.1  Number of Samples K for Expectation Estimation

| K | Percentiles↓ Mean | % Below↑ 2Å | 5 Å |
|---|---|---|---|
| 5 | 7.5 | 37.1 | 62.7 |
| 10 | 7.0 | 39.5 | 65.3 |
| 20 | 6.3 | 41.2 | 71.3 |
| 40 | 5.3 | 42.3 | 72.5 |
| 60 | 5.2 | 42.9 | 73.2 |
| 80 | **5.2** | **43.0** | **73.5** |

Table 3: Flexible blind Top-1 self-docking performance. Comparative results on the number of samples $K$ for expectation estimation.

In this part, for computation efficiency, the model is trained only for 100 epochs. By fixing $k = \frac{1}{10}$ and $\eta = 1$, we train the force model by setting $K \in \{5, 10, 20, 40, 60, 80\}$. The results are shown in Table 3. The metrics evaluated include the mean RMSD and the percentages below specific thresholds. While $K = 80$ achieves the best performance across all metrics, the improvements between $K = 40$ and 80 are negligible compared to the computational cost of increasing $K$. Considering the balance between computational efficiency and model performance, $K = 40$ is selected as the optimal choice. This value provides a good trade-off, offering near-peak performance with significantly reduced computation compared to $K = 80$.

## E.2  Guidance Strength

In this part, by fixing $k = \frac{1}{10}$ and $K = 40$, we evaluate the effect of $\eta$ by setting $\eta = 0.1$ to 1.2 with step 0.1. The results are presented in Figure 3. It can be observe that as $\eta$ increases up to 1.0, both the vina score and mean RMSD decrease, the percentage of ligand atomic RMSD less than 2Å, as well as the proportion of ligands passing the PoseBuster tests, improve. Thus, we choose $\eta = 1$.

## E.3  Role of P2Rank for Pocket Identification

As noted in prior work [11, 8, 12], predicting binding pockets can significantly reduce the conformational search space and improve the efficiency and accuracy of docking models. In this work, we adopt P2Rank to estimate the binding site center and initialize the ligand around it—this aligns with standard practice in blind docking evaluation.

To ensure a fair comparison and factor out the influence of P2Rank, we conducted additional experiments. Since FABind [8] and FABind+ [12] incorporate built-in pocket prediction modules, we added a DiffDock + P2Rank baseline for direct comparison. As shown in the table 4, integrating P2Rank improves the mean RMSD and centroid distance significantly, with minimal change in the %RMSD < 2Åmetric—indicating more precise pose sampling near the true binding site, even if the top-ranked pose isn't always within the strict 2Åthreshold. More importantly, we find that our method is robust to small perturbations in the predicted pocket location. To evaluate sensitivity, we injected Gaussian noise with increasing variance ($\sigma = 1$ to 10 Å) into the P2Rank-predicted pocket center

Table 4: Effect of pocket prediction (P2Rank) on DiffDock (Top-1, PDBBind).

| Method | Mean RMSD ↓ | % RMSD≤2Å ↑ | Mean Centroid ↓ | % Centroid≤2Å ↑ |
|---|---|---|---|---|
| DiffDock | 7.5 | 38.2 | 5.5 | 60.8 |
| DiffDock + P2Rank | 5.4 | 37.6 | 3.5 | 67.3 |

before ligand initialization. As shown below, performance remains stable across all noise levels up to $\sigma = 5$ Å, with only minor degradation at $\sigma = 10$ Å, comparable to results without P2Rank at all.

Table 5: Sensitivity to pocket-center noise (Gaussian, $\sigma$ Å).

| Method / Setting | Mean RMSD ↓ | % RMSD≤2Å ↑ | Mean Centroid ↓ | % Centroid≤2Å ↑ |
|---|---|---|---|---|
| Ours (no noise) | 4.2 | 41.1 | 3.1 | 71.7 |
| $\sigma=1$ | 4.2 | 41.2 | 3.0 | 72.3 |
| $\sigma=2$ | 4.5 | 41.7 | 3.1 | 71.4 |
| $\sigma=3$ | 4.3 | 41.5 | 3.1 | 71.7 |
| $\sigma=5$ | 4.4 | 41.5 | 3.2 | 70.9 |
| $\sigma=10$ | 4.8 | 39.8 | 3.8 | 66.1 |
| w/o P2Rank | 4.8 | 39.4 | 3.7 | 66.9 |

## E.4   Post-Minimization Ablation

A critical question whether the improvements from our force-guided framework stem from genuine integration during generation or could be replicated by post-hoc energy minimization of poses generated by a standard model. To this end, we conducted a additional ablation experiment, where we took the top-1 poses generated by both DiffDock (w or w/o force model) and our baseline model ("Ours"), and applied energy minimization using AutoDock Vina's local optimizer (without re-docking). The results are summarized in the table 6. It can be observed that:

- Energy minimization dramatically improves PoseBuster scores, increasing them from 14–24% to over 40%. This confirms that poor physical plausibility is largely due to local geometric distortions (e.g., clashes, bad bond angles) that minimization can fix.

- However, minimization has limited impact on RMSD — it slightly reduces mean RMSD but does not significantly improve the percentage of poses below 2Å. In some cases, RMSD even increases slightly, likely because the minimizer pulls the ligand away from the crystal pose while optimizing internal energy.

- Combining force-guided generation with minimization yields the best of both worlds: 45.8% PoseBuster pass rate, significantly outperforming any other combination.

Table 6: Post-hoc Vina minimization on model variants (Top-1, PDBBind).

| Method | Mean RMSD ↓ | % RMSD≤2Å ↑ | PoseBuster ↑ |
|---|---|---|---|
| DiffDock | 7.5 | 38.2 | 14.3 |
| DiffDock + Mini | 7.0 | 37.9 | 40.3 |
| DiffDock + Vina | 3.8 | 43.5 | 24.6 |
| DiffDock + Vina + Mini | 3.9 | 42.8 | 42.5 |
| Ours | 1.7 | 41.1 | 17.5 |
| Ours + Mini | 1.8 | 40.8 | 43.1 |
| Ours + Vina | 1.1 | 48.6 | 29.7 |
| Ours + Vina + Mini | 1.2 | 46.2 | 45.8 |

## E.5 GNINA scoring function

We have conducted additional experiments on PDBBind testset using CNNscore and CNNaffinity as the guiding potentials in our force-guided framework. The results are summarized below. This confirms that our method is not only compatible with classical physics-based scoring but also benefits from data-driven, deep learning-based energy models that capture complex protein-ligand interaction patterns.

Table 7: Performance of the force-guided framework using different scoring functions for GNINA.

| Method | Mean RMSD ↓ | % RMSD≤2Å ↑ | Mean Centroid ↓ | % Centroid≤2Å ↑ |
|---|---|---|---|---|
| GNINA + Affinity | 3.1 | 47.5 | 2.4 | 74.3 |
| GNINA + CNNscore | 3.2 | 46.3 | 2.5 | 72.9 |
| GNINA + CNNaffinity | 3.1 | 48.6 | 2.3 | 75.8 |

## E.6 DockGen Cross-Domain Benchmark

For the generalization assessment, stronger OOD evaluation is desirable. To better assess the generalizability of our method in a realistic cross-docking scenario, we evaluate our model—trained solely on the PDBBind dataset—on the DockGen test set. This setup provides a test of cross-domain generalization. The results are shown as follows in table 8. Our results show that when integrating our plug-and-play force guidance module with the baseline model, there is a significant improvement compared to the baseline model on this challenging OOD benchmark. This demonstrates that our method not only enhances docking accuracy but also improves robustness to structural and functional shifts across the proteome.

Table 8: DockGen cross-domain evaluation (train: PDBBind only).

| Method | Mean RMSD ↓ | % RMSD≤2Å ↑ | Mean Centroid ↓ | % Centroid≤2Å ↑ |
|---|---|---|---|---|
| FABind | 19.1 | 1.3 | 18.1 | 14.2 |
| FABind+ | 18.5 | 1.5 | 16.9 | 16.1 |
| DiffDock | 16.2 | 5.3 | 14.5 | 21.4 |
| DiffDock + Vina | 12.5 | 7.1 | 10.9 | 25.7 |
| DiffDock + Conf | 13.4 | 6.9 | 12.5 | 24.6 |
| DiffDock + Gnina | 12.9 | 7.2 | 11.2 | 25.3 |
| DiffDock + Glide | 12.6 | 7.2 | 11.9 | 26.6 |
| Ours | 14.5 | 6.5 | 14.1 | 23.2 |
| Ours + Vina | 12.3 | 8.1 | 10.4 | 26.8 |
| Ours + Conf | 12.5 | 7.9 | 11.6 | 25.4 |
| Ours + Gnina | 13.1 | 7.6 | 12.1 | 26.9 |
| Ours + Glide | **12.2** | **8.1** | **10.3** | **27.8** |

## E.7 Visual Quality Assessment

In Figure 4, we present a comparative visual analysis involving our model with Vina force guidance (Red), FABind+ (Purple), and the actual Crystal Structure (Green) of ligands for PDBIDs 6oy1 and 6uvv. This side-by-side comparison highlights the positioning accuracy of our model relative to FABind+.

Notably, although FABind+ achieves similar RMSD performance for 6oy1, the energy of the generated ligand is extremely high, as indicated by the Vina score. This high energy suggests that the generated conformation is chemically unstable, a common issue with many DL-based methods that lack sufficient physical guidance. In contrast, the ligand generated by our force-guided model has a much lower Vina score, indicating a more stable and physically plausible conformation. As for 6uvv,

FABind+ generates an incorrect ligand conformation, which further emphasizes the robustness of our method.

In summary, our model not only demonstrates superior RMSD performance but also generates ligands that are chemically stable and biologically relevant. These ligands accurately replicate crucial hydrogen bond interactions with protein residues. This visual assessment clearly highlights the enhanced precision and adaptability of our model in generating ligands with physical plausibility.

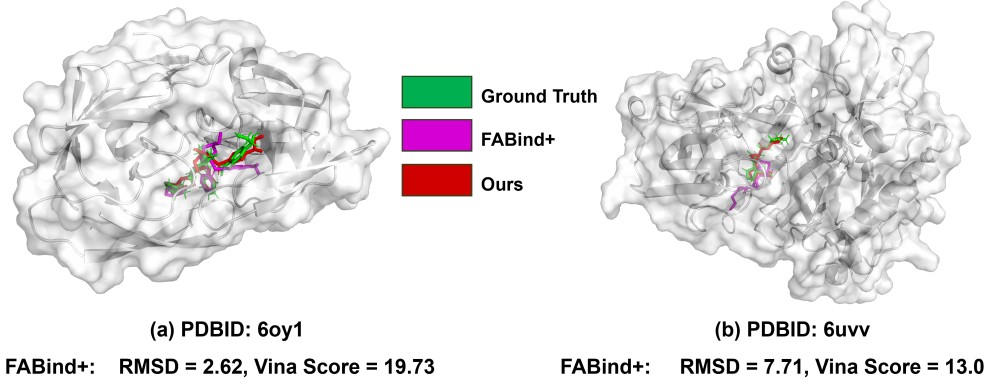

| | (a) PDBID: 6oy1 | | (b) PDBID: 6uvv | |
|---|---|---|---|---|
| FABind+: | RMSD = 2.62, Vina Score = 19.73 | FABind+: | RMSD = 7.71, Vina Score = 13.0 |
| ForceFM: | RMSD = 1.62, Vina Score = -6.64 | ForceFM: | RMSD = 0.72, Vina Score = - 5.43 |

Figure 4: Comparative visualization of generated ligands for PDBIDs 6oy1 and 6uvv, displayed from left to right. The color coding is as follows: Our model with force guidance (Red), FABind+ (Purple), and the actual Crystal Structure (Green). RMSDs and Vina scores are provided, highlighting the precision and accuracy of our method in generating low-energy ligand conformations with improved physical plausibility.

## F   Implementing Detail for Training/Inference DiffDock with Energy Guidance

First of all, it needs to point out that the time index for prior and data distribution for diffusion model is 1 and 0 while flow matching perform it in the reverse way.

Let $p_0 (\mathbf{x}_0 \mid \mathbf{y}) = q_0 (\mathbf{x}_0 \mid \mathbf{y}) \frac{e^{-k\mathcal{E}_0(\mathbf{x}_0, \mathbf{y})}}{Z}$. We are able to sampling from $p_0$ by the reverse-time SDE with the score function:

$$\nabla_{\mathbf{x}_t} \log p_t (\mathbf{x}_t, \mathbf{y}) = s_\theta (\mathbf{x}_t, \mathbf{y}, t) - k\nabla_{\mathbf{x}_t}\mathcal{E}_t (\mathbf{x}_t, \mathbf{y}) \tag{34}$$

where $s_\theta (\mathbf{x}_t, \mathbf{y}, t)$ denotes the given score model, $\nabla_{\mathbf{x}_t}\mathcal{E}_t (\mathbf{x}_t, \mathbf{y})$ is the intermediate force guidance. Lemma A.1 provides the precise force function at time $t$, thereby advancing our understanding of the intermediate energy function. Thus, similar to ForceFM, an additional force network is proposed to approximate $\nabla_{\mathbf{x}_t}\mathcal{E}_t (\mathbf{x}_t, \mathbf{y})$.

**Training process.** We first employ the given score model to generate ligand candidates from $q_0 (\mathbf{x}_0 \mid \mathbf{y})$, which are then used to train an independent intermediate force network $h_\psi(\mathbf{x}_t, \mathbf{y}, t)$. For each protein-ligand pair $(\mathbf{x}_0, \mathbf{y})$, we first sample $K$ ligand conformations $\mathbf{x}_0^{(1:K)} \sim q_0 (\mathbf{x}_0 \mid \mathbf{y})$ and perturb the data at time $t$ according to the forward SDE. We define the intermediate force loss function $L_{force}(\psi)$ as

$$\mathbb{E}_{p(t)}\mathbb{E}_{(\mathbf{x}_0, y)} \left[ ||h_\psi(\mathbf{x}_t, \mathbf{y}, t) - \sum_{i=1}^{K} w_i \zeta \left( \mathbf{x}_0^{(i)}, \mathbf{x}_t, \mathbf{y} \right) ||_2^2 \right], \tag{35}$$

where $w_i = \frac{q_t \left( \mathbf{x}_t | \mathbf{x}_0^{(i)}, \mathbf{y} \right) e^{-k\mathcal{E}_0 \left( \mathbf{x}_0^{(i)}, \mathbf{y} \right)}}{\sum_{j=1}^{K} q_t \left( \mathbf{x}_t | \mathbf{x}_0^{(j)}, \mathbf{y} \right) e^{-k\mathcal{E}_0 \left( \mathbf{x}_0^{(j)}, \mathbf{y} \right)}}$ and $\zeta \left( \mathbf{x}_0^{(i)}, \mathbf{x}_t, \mathbf{y} \right) = \nabla_{\mathbf{x}_t} \log q_t (\mathbf{x}_t \mid \mathbf{y}) -$ $\nabla_{\mathbf{x}_t} \log q_t \left( \mathbf{x}_t \mid \mathbf{x}_0^{(i)}, \mathbf{y} \right)$. The latter component of Eqn.(35) signifies the precise value of the intermediate force at time $t$, where $\nabla_{\mathbf{x}_t} \log q_t (\mathbf{x}_t \mid \mathbf{y})$ is the estimated by the given score model

**Algorithm 2:** Training Procedure (Single Epoch) for Force Model

---

**Input :** Training pairs $\{(\mathbf{x}_0, \mathbf{y})\}$, score model $s_\theta(\mathbf{x}_t, y, t)$, energy $\mathcal{E}_0(\cdot)$, intermediate force
network $h_\psi(\mathbf{x}_t, y, t)$

**for** *each* $\mathbf{x}_0, \mathbf{y}$ **do**

  $t \sim \mathcal{U}(0, 1);$

  $\mathbf{x}_t \sim q_t(\mathbf{x}_t \mid \mathbf{x}_0);$ # perturb data

  $\mathbf{x}_0^{(1:K)} \sim q_0(\mathbf{x}_0 \mid y);$

  **for** $i = 1, ..., K$ **do**

  $\quad \Big\lfloor \; \zeta\left(\mathbf{x}_0^{(i)}, \mathbf{x}_t, \mathbf{y}\right) = s_\theta(\mathbf{x}_t, \mathbf{y}, t) - \nabla \log q_t\left(\mathbf{x}_t \mid \mathbf{x}_0^{(i)}, \mathbf{y}\right);$

  $Y = \dfrac{\sum_i \zeta\left(\mathbf{x}_0^{(i)}, \mathbf{x}_t, \mathbf{y}\right) q_t\left(\mathbf{x}_t \mid \mathbf{x}_0^{(i)}, \mathbf{y}\right) e^{-k\mathcal{E}_0\left(\mathbf{x}_0^{(i)}, \mathbf{y}\right)}}{\sum_i q_t\left(\mathbf{x}_t \mid \mathbf{x}_0^{(i)}, \mathbf{y}\right) e^{-k\mathcal{E}_0\left(\mathbf{x}_0^{(i)}, \mathbf{y}\right)}};$

  $\mathcal{L} = \|h_\psi(\mathbf{x}_t, y, t) - Y\|_2^2;$

  $\min_\psi \mathcal{L}.$

---

$s_\theta(\mathbf{x}_t, \mathbf{y}, t)$ and $\nabla_{\mathbf{x}_t} \log q_t\left(\mathbf{x}_t \mid \mathbf{x}_0^{(i)}, \mathbf{y}\right)$ is the tractable perturbation kernel on the product space defined by transition, rotation and torsion. The force-guided training process is summarized in Algorithm 2

