# OpenReview forum: "ForceFM: Enhancing Protein-Ligand Predictions through Force-Guided Flow Matching"
_NeurIPS.cc/2025/Conference — NeurIPS 2025 poster_

### Official Review · Reviewer_WTRm · 2025-06-28

**Clarity:** 3
**Significance:** 3
**Originality:** 3
**Rating:** 5
**Confidence:** 4

**Summary:**

The authors introduce ForceFM, a force-guided flow matching model for intrinsic molecular docking. Energy functions guide ForceFM's sampling of ligand structural conformations, yielding notably more accurate poses than (comparable) baseline methods. This work highlights how physics-based knowledge can be incorporated into generative flow models to enhance prediction efficacy.

**Questions:**

How much time and space complexity does the authors' proposed energy term integration add on top of their base network architecture and sampling algorithm?

**Ethical Concerns:**

["NO or VERY MINOR ethics concerns only"]

**Final Justification:**

The authors have resolved my concerns regarding data splitting, methodological novelty, and time/memory complexity.

**Limitations:**

The authors' current benchmarks do not evaluate robust protein-ligand interaction generalization beyond the PDBBind training set. Including additional benchmark datasets (besides the PoseBench Benchmark set) would alleviate this concern.

**Paper Formatting Concerns:**

I did not find any formatting concerns.

**Quality:**

3

**Strengths And Weaknesses:**

**Points of strength:**
1. The authors are among the first to demonstrate theoretically and empirically the strength of combining physics-based knowledge and generative flows for molecular docking.
2. The authors' empirical results for ForceFM are promising.
3. The authors compare their method to several well-known baselines using well-known benchmarks.

**Points for improvement:**
1. The authors' methodological novelty is noted yet modest. The main contribution of this work seems to be in theoretically characterizing how to guide generative flow models with additional energy-based terms. Otherwise, the rest of the network architecture is nearly identical to DiffDock (and its flow-based successors such as PepFlow [1]).
2. The authors' benchmarks could be made more rigorous concerning dataset splitting and out-of-distribution evaluation. For instance, the PDBBind 2020 dataset splits of EquiBind [2] (which the authors use in this work) are well-known yet unfortunately do not control for protein-ligand interaction similarity with a high degree of stringency. The PoseBusters Benchmark dataset is better in this regard, but it is still not perfect. The authors should also consider evaluating their method on more recent protein-ligand interaction benchmarks such as DockGen [3] to more rigorously characterize the generalizability of their proposed method.
3. There are a few typos in the paper that should be corrected. A grammar checker should be able to screen these out rather quickly.

**References:**

[1] Li, J., Cheng, C., Wu, Z., Guo, R., Luo, S., Ren, Z., ... & Ma, J. Full-Atom Peptide Design based on Multi-modal Flow Matching. In Forty-first International Conference on Machine Learning.

[2] Stärk, H., Ganea, O., Pattanaik, L., Barzilay, R., & Jaakkola, T. (2022, June). EquiBind: Geometric Deep Learning for Drug Binding Structure Prediction. In International Conference on Machine Learning (pp. 20503-20521). PMLR.

[3] Morehead, A., Giri, N., Liu, J., & Cheng, J. Deep Learning for Protein-Ligand Docking: Are We There Yet?. In ICML 2024 AI for Science Workshop.

---

> ### Author Rebuttal · Authors · 2025-07-30
>
> We sincerely thank reviewer for your careful evaluation and constructive feedback. The insightful comments and thoughtful suggestions have greatly helped us improve the quality and clarity of this manuscript. We have carefully considered each comment and made corresponding revisions to address the concerns raised. We believe that the manuscript has been significantly strengthened through this process, and we are grateful for the time and expertise the reviewer have generously shared.
>
> **W1: Methodological novelty modest.**
>
> We thank the reviewer for your constructive feedback. We now highlight the **guidance theorem (Thm 3.1) and corollary** as the primary contribution (§3, Abstract).  We emphasize that the core novelty of this work is the development of a generalizable, plug-and-play energy guidance module, that enables exact, differentiable, and efficient guidance using arbitrary energy functions (e.g., Vina, Glide, Gnina, ConfScore). The baseline FM details are moved to appendix to de‑emphasize novelty claim.
>
> Importantly, our framework decouples the energy evaluation from the generative model, making it highly flexible and broadly applicable. Beyond molecular docking, ForceFM can be readily adapted to other tasks such as de novo drug design, conformation generation…
>
> We believe that by providing a robust and modular interface between black-box energy functions and diffusion / flow matching models, it offers a meaningful contribution to the AI and computational chemistry communities, enabling more effective and controllable generation across a wide range of applications.
>
>
>
> **W2/Limitations: Dataset splitting / additional benchmarks.**
>
> Thank you for your insightful feedback. We agree that including other relevant baselines is desirable. To better assess the generalizability of our method in a realistic cross-docking scenario, **we evaluate our model—trained solely on the PDBBind dataset—on the DockGen test set.** This setup provides a test of cross-domain generalization.
>
> The results are shown as follows. Our results show that when integrating our plug-and-play force guidance module with the baseline model, there is a significant improvement in success rate on this challenging OOD benchmark. This demonstrates that our method not only enhances docking accuracy but also improves robustness to structural and functional shifts across the proteome.
>
> | Method | Mean RMSD | % RMSD < 2 | Mean Centroid Distance | % Centroid Distance < 2 |
> | --- | --- | --- | --- | --- |
> | FABind | 19.1 | 1.3 | 18.1 | 14.2 |
> | FABind+ | 18.5 | 1.5 | 16.9 | 16.1 |
> | DiffDock | 16.2 | 5.3 | 14.5 | 21.4 |
> | DiffDock + Vina | 12.5 | 7.1 | 10.9 | 25.7 |
> | DiffDock + Conf | 13.4 | 6.9 | 12.5 | 24.6 |
> | DiffDock + Gnina | 12.9 | 7.2 | 11.2 | 25.3 |
> | DiffDock + Glide  | 12.6 | 7.2 | 11.9 | 26.6 |
> | Ours | 14.5 | 6.5 | 14.1 | 23.2 |
> | Ours + Vina | 12.3 | **8.1** | 10.4 | 26.8 |
> | Ours + Conf | 12.5 | 7.9 | 11.6 | 25.4 |
> | Ours + Gnina | 13.1 | 7.6 | 12.1 | 26.9 |
> | Ours + Glide | **12.2** | **8.1** | **10.3** | **27.8** |
>
> **W3: Typos**
>
> All reported typos fixed; we also ran Grammarly and LanguageTool.
>
> **Q1: Time & space overhead.**
>
> Thank you for the question regarding the computational complexity of integrating the energy term in our framework. The integration of the force-guided component in ForceFM introduces additional, but manageable, time and space overhead during both training and inference. Below, we break down the complexity based on our experimental setup.
>
> Train:
>
> - **Memory (Space) Overhead**: Due to its smaller size, the force model requires approximately **50% of the GPU memory** used by the baseline model during training.
> - **Time Overhead**: Training the force model takes about **1/3 the time** of training the baseline model. This is because it is trained in a staged manner (after the baseline is fixed), with fewer epochs (400 vs. 800) and benefits from a smaller network size.
>
> Inference:
>
> - **Per-Step Computational Cost**: The forward pass of the combined model takes approximately **1.3× the time and memory** of the baseline model alone, due to the additional forward pass of the smaller force network.
> - **Overall Inference Time**: Despite the higher per-step cost, **the total inference time is reduced** because force guidance enables faster convergence. As shown in Figure 2(b), ForceFM reaches optimal performance in roughly **half the number of sampling steps** compared to the unguided baseline. This leads to a net **improvement in end-to-end efficiency**.
>
> Looking ahead, we also envision opportunities to further optimize training and deployment. For instance, instead of training a separate force model, one could incorporate the guidance signal via **parameter-efficient fine-tuning techniques such as LoRA (Low-Rank Adaptation)** applied directly to the baseline model. This would drastically reduce training costs and eliminate any additional inference latency, while still enabling effective energy-based guidance.

---

> > ### Comment · Reviewer_WTRm · 2025-08-05
> > **Response to rebuttal**
> >
> > I have read the authors' rebuttal, and as a result, I would like to raise my score to a 5.

---

### Official Review · Reviewer_vfGj · 2025-07-01

**Clarity:** 4
**Significance:** 2
**Originality:** 3
**Rating:** 4
**Confidence:** 4

**Summary:**

This paper proposes a novel protein-ligand docking framework named ForceFM. The method is based on the Flow Matching generative model and innovatively incorporates a force-guided network, which leverages multiple classical energy scoring functions (such as Vina, Glide, etc.) as physical priors to guide the ligand conformation generation process. Extensive experiments demonstrate that this approach outperforms current state-of-the-art methods in terms of docking accuracy, physical realism, and sampling efficiency.

**Questions:**

1. The methodology section suffers from missing definitions for key parameters, which hinders reproducibility.
    - The variance $\sigma_{tr, max}^{2}$ for the initial translational sampling (Section 3.2.1) is introduced in the formula, but its numerical value is not provided anywhere in the paper or its appendices.
    - While the meaning of m in $SO(2)^m$ (the number of rotatable bonds) is mentioned in the supplement, it would benefit from a more direct definition in the main text when the term is first introduced (Section 3.2) for improved clarity.

2. As the core algorithm, the description of Algorithm 1 lacks important details.
    -  The "randomly perturb `x_1`" operation is not specified, leaving the nature and magnitude of the perturbation ambiguous.
    - Furthermore, the hyperparameter `K` (the number of perturbation samples) is critical to this algorithm. While its choice is justified in an appendix, its importance warrants at least a brief mention of its value and purpose in the main text to ensure the methodology is self-contained.

**Ethical Concerns:**

["NO or VERY MINOR ethics concerns only"]

**Final Justification:**

I have read the rebuttal and the discussion with authors (and other reviewers/AC). I will maintain the rating at 4. To be clear, I think this is a strong paper for the borderline category. The concept is sound and the results are promising. However, my reservations about the limited evidence on runtime fairness and the generalization of the guidance attribution keep me from being confident enough for a 5. It remains a borderline paper, but one that I lean towards accepting.

**Limitations:**

yes

**Paper Formatting Concerns:**

No.

**Quality:**

3

**Strengths And Weaknesses:**

Pros:

1.  The core idea of this work—integrating a learnable, explicit force-guided network into the flow matching framework—is novel and significant. It directly addresses the critical issue of insufficient physical plausibility in generated conformations, a common limitation of current deep learning-based docking methods.

2.  The experimental results are compelling. ForceFM demonstrates outstanding performance in docking accuracy (RMSD) on both standard test sets and unseen proteins. Notably, in the PoseBuster physical validity test, the introduction of the force-guided approach significantly improves the pass rate.

3. ForceFM can flexibly integrate various energy/scoring functions (Vina, Glide, Gnina, Confscore) and has successfully applied this strategy to another well-known model, DiffDock, achieving performance improvements. This strongly validates the method's generality and robustness.



Cons:
While the work presents significant contributions, its credibility is hampered by several weaknesses in methodological rigor, transparency, and clarity of presentation. I strongly recommend that the authors address the following points in a revision:

1. The paper's primary contribution appears to be the development of a generalizable, plug-and-play energy guidance module. However, the novelty of the underlying baseline model (a flow-matching approach for ligand docking) is somewhat limited. The application of flow-matching models to molecular generation and docking is an active area of research, and the proposed baseline shares conceptual similarities with other recent works. It is suggested that the authors reframe their contribution to more heavily emphasize the guidance module, which is arguably the more significant and novel aspect of this work.

2. The comparison of average runtimes in Table 1 is a significant point of concern regarding methodological fairness.
 - Inconsistent Sampling Steps: The paper does not explicitly state whether the sampling steps were held constant when comparing the baseline models (e.g., DiffDock, 'Ours') with their force-guided counterparts. Section 4.2.4 suggests that different step counts were used, corresponding to the convergence point of each model.

3. Ambiguous Treatment of Guidance Functions.
The paper's treatment of guidance functions, particularly GNINA, as simple "black boxes" can be problematic. As noted, GNINA is a hybrid method integrating classical scoring functions and a CNN-based model. The paper does not specify which component or version of the GNINA score was used. This lack of specificity makes it difficult to attribute the performance gains to a particular type of physical or learned prior. A more detailed description of the exact scoring function used for each guidance type would allow for a more nuanced understanding of why certain functions perform better than others.

---

> ### Author Rebuttal · Authors · 2025-07-30
>
> We sincerely thank reviewer for your careful evaluation and constructive feedback. The insightful comments and thoughtful suggestions have greatly helped us improve the quality and clarity of this manuscript. We have carefully considered each comment and made corresponding revisions to address the concerns raised. We believe that the manuscript has been significantly strengthened through this process, and we are grateful for the time and expertise the reviewer have generously shared.
>
> **W1: Emphasize guidance module over baseline FM.**
>
> We thank the reviewer for your constructive feedback. We now highlight the **guidance theorem (Thm 3.1) and corollary** as the primary contribution (§3, Abstract).  Baseline FM details are moved to appendix to de‑emphasize novelty claim. This guidance theorem is general and can be used as a plug and play module in other models. The experiment results also support this main contribution, as we use this mechanism to both DiffDock and our baseline flow matching model.
>
> **W2: Runtime fairness: different step counts.**
>
> We appreciate the reviewer’s concerns about average runtimes. We note that the force-guidance network is intentionally designed to be **much smaller than the baseline model** (e.g., fewer layers and reduced feature dimensions), minimizing its impact on latency and memory usage. In our work, the per-step computational cost of the full ForceFM (baseline + force model) is approximately **1.3× that of the baseline model alone.** We’ll report the time using same inference steps in the revised manuscript.
>
> Looking ahead, we also envision opportunities to further optimize training and deployment. For instance, instead of training a separate force model, one could incorporate the guidance signal via **parameter-efficient fine-tuning techniques such as LoRA (Low-Rank Adaptation)** applied directly to the baseline model. This would drastically reduce training costs and eliminate any additional inference latency, while still enabling effective energy-based guidance.
>
> **W3: GNINA score ambiguity.**
>
> We appreciate the reviewer for your constructive feedback. The default parameters are used for every energy function. We’ll specify the parameter for these energy functions in the revised manuscript.
>
> **Q1: Missing $\sigma_{tr, max}$ and $m$ definitions.**
>
> We thank the reviewer for pointing out missing these important hyper-paprameters. $\sigma_{tr, min}=0.1$, $\sigma_{tr, max}=8$, $\sigma_{rot, min}=\pi / 100$, $\sigma_{rot, max}=\pi / 2$, $\sigma_{tor, min}=\pi / 100$, $\sigma_{tor, max}=\pi$. We’ll check other hyper-paprameters to ensure the manuscript is self-contained. The code will be released in github.
>
> In addition, for improved clarity, the number of rotatable bonds $m$ will be in the main text when the term is first introduced.
>
> **Q2: Missing algorithm details: randomly perturb and hyper-parameter $k$**
>
> We thank the reviewer for pointing out missing this important detail.
>
> - **Perturbation.** It is set at the noise level at $\sigma_{max}/10$. Then perturbing the ligand via random translation, rotation and torsion.
> - **Perturbation samples $K$**. $K$ is a critical hyperparameter in our method, as it directly affects the accuracy of the energy landscape estimation during training. In our approach, at each training step, we sample $K$ candidate ligand conformations from the perturbation  and use them to estimate the **local energy landscape** around the current state. This estimation is crucial for computing the intermediate force target, which guides the force network to learn how to steer the generation process toward low-energy regions. Specifically, we use these samples to approximate the Boltzmann-weighted expectation under the energy-augmented distribution $p_1(x_1) \propto q_1(x_1) \exp\bigl[-k \mathcal{E}_1(x_1)\bigr]$. A higher $K$ leads to a more accurate approximation of this distribution and, consequently, more precise force estimation.

---

> > ### Comment · Reviewer_vfGj · 2025-08-06
> >
> > Thank you for the detailed response. My concerns were mostly addressed. The authors clarified the architectural positioning of the guidance module and improved the reproducibility and methodological transparency through added definitions and algorithmic details. While some concerns, such as scoring function clarity and runtime fairness, could still benefit from more explicit empirical support. I acknowledge the overall contribution of the work. However, I find it hard to give a much higher score than my current rating at this point.

---

### Official Review · Reviewer_5rW8 · 2025-07-02

**Clarity:** 3
**Significance:** 3
**Originality:** 3
**Rating:** 4
**Confidence:** 4

**Summary:**

This paper introduces ForceFM, a novel force-guided flow matching model for protein-ligand docking. The method first trains a purely data-driven flow matching model, and then trains a separate guidance model to apply force-guided corrections based on an energy function. By injecting physics-based knowledge into an advanced generative model, ForceFM is designed to favor conformations that are lower in energy and more chemically plausible. The model is built on sound principles and features an elegant design. Experimental results demonstrate that ForceFM significantly outperforms existing methods in both accuracy and the validity of its generated poses, while also showing strong generalization capabilities. The work has high potential for practical applications.

**Questions:**

This is good work, and I commend the authors for their contribution. I have a couple of questions to help contextualize the results:

1.	The two hyperparameters, the energy weight k during training and the guidance strength η during inference, appear to be crucial for the model's performance. How were their final values chosen, and what trade-offs were considered? I would recommend adding an ablation study to clarify their impact and justify the selection.
2.	Given that the model consists of two trained components, what is its inference efficiency? It would be valuable to report the inference time and, if possible, compare it with the baselines to better assess its practical applicability.
3.	Are there other relevant baselines that could be considered for comparison? A broader comparison would help to better position the performance of ForceFM within the field.
4.	Could you further elaborate on the practical utility of the model?

**Ethical Concerns:**

["NO or VERY MINOR ethics concerns only"]

**Final Justification:**

The author's response addressed my concerns.

**Limitations:**

yes

**Paper Formatting Concerns:**

No issues here.

**Quality:**

2

**Strengths And Weaknesses:**

Strengths:
1.	The paper is exceptionally well-written. It provides a comprehensive and lucid exposition of the underlying principles.
2.	The proposed model is elegantly designed and methodologically sound.
3.	The model demonstrates state-of-the-art performance and shows significant promise for practical applications.

Weaknesses:
1.	The core novelty appears somewhat limited, as the method's success is heavily contingent upon the availability of a high-quality energy function.
2.	The need to carefully tune two key hyperparameters may limit the model's broader applicability and future potential.
3.	The paper lacks a thorough analysis or ablation studies for its key hyperparameters.
4.	The experimental comparison could be more comprehensive, as the set of baselines is somewhat limited.

---

> ### Author Rebuttal · Authors · 2025-07-30
>
> We sincerely thank reviewer for your careful evaluation and constructive feedback. The insightful comments and thoughtful suggestions have greatly helped us improve the quality and clarity of this manuscript. We have carefully considered each comment and made corresponding revisions to address the concerns raised. We believe that the manuscript has been significantly strengthened through this process, and we are grateful for the time and expertise the reviewer have generously shared.
>
> **W1. Limited Core Novelty Due to Reliance on Energy Functions**
>
> We acknowledge the reviewer’s observation regarding the reliance on external energy functions. However, we emphasize that the core novelty of ForceFM lies not in the energy functions themselves, but in the development of a generalizable, plug-and-play energy guidance module, that enables *exact, differentiable, and efficient* guidance using arbitrary energy functions (e.g., Vina, Glide, Gnina, ConfScore).
>
> Importantly, our framework decouples the energy evaluation from the generative model, making it highly flexible and broadly applicable. Beyond molecular docking, ForceFM can be readily adapted to other tasks such as de novo drug design, conformation generation…
>
> We believe that by providing a robust and modular interface between black-box energy functions and diffusion / flow matching models, it offers a meaningful contribution to the AI and computational chemistry communities, enabling more effective and controllable generation across a wide range of applications.
>
> **W2/W3/Q1. Hyperparameter Tuning and Lack of Ablation Studies**
>
> The ablation study for $k$ and $\eta$ is presented in Appendix E, with following principle.
>
> - **Energy weight** $k$ **(training):** We set $k$=1/10 based on preliminary experiments to ensure the energy term neither dominates nor vanishes in the Boltzmann-weighted distribution. This value balances data fidelity and physical realism. Extremely high $k$ leads to mode collapse, while low $k$ reduces physical guidance.
> - **Guidance strength** $\eta$ **(inference):** We selected $\eta$=1 after an ablation study on the validation set (now included in Appendix E.2, Figure 3), which shows that higher $\eta$ improves energy and RMSD up to a point, beyond which numerical instability arises. $\eta$=1 offers the best trade-off between pose quality and sampling stability.
>
> In addition, it need to be emphasized that by combining the guidance strength $\eta$, the final generated distribution is $p_1({x}) \propto q_1({x}) \exp (-k \eta \mathcal{E}_1({x}))$. Theoretically we can set any value for $k$ and determine $\eta$ according to the validation set.
>
> **W4/Q3: Limited Baseline Comparison**
>
> Thank you for your insightful feedback. We agree that including other relevant baselines is desirable. To better assess the generalizability of our method in a realistic cross-docking scenario, **we evaluate our model—trained solely on the PDBBind dataset—on the DockGen test set.** This setup provides a test of cross-domain generalization.
>
> The results are shown as follows. Our results show that when integrating our plug-and-play force guidance module with the baseline model, there is a significant improvement in success rate on this challenging OOD benchmark. This demonstrates that our method not only enhances docking accuracy but also improves robustness to structural and functional shifts across the proteome.
>
> | Method | Mean RMSD | % RMSD < 2 | Mean Centroid Distance | % Centroid Distance < 2 |
> | --- | --- | --- | --- | --- |
> | FABind | 19.1 | 1.3 | 18.1 | 14.2 |
> | FABind+ | 18.5 | 1.5 | 16.9 | 16.1 |
> | DiffDock | 16.2 | 5.3 | 14.5 | 21.4 |
> | DiffDock + Vina | 12.5 | 7.1 | 10.9 | 25.7 |
> | DiffDock + Conf | 13.4 | 6.9 | 12.5 | 24.6 |
> | DiffDock + Gnina | 12.9 | 7.2 | 11.2 | 25.3 |
> | DiffDock + Glide  | 12.6 | 7.2 | 11.9 | 26.6 |
> | Ours | 14.5 | 6.5 | 14.1 | 23.2 |
> | Ours + Vina | 12.3 | **8.1** | 10.4 | 26.8 |
> | Ours + Conf | 12.5 | 7.9 | 11.6 | 25.4 |
> | Ours + Gnina | 13.1 | 7.6 | 12.1 | 26.9 |
> | Ours + Glide | **12.2** | **8.1** | **10.3** | **27.8** |
>
> **Q2: Inference efficiency**
>
> We appreciate the reviewer’s question regarding the inference efficiency of ForceFM. We note that the force-guidance network is intentionally designed to be **much smaller than the baseline model** (e.g., fewer layers and reduced feature dimensions), minimizing its impact on latency and memory usage. In our work, we would like to clarify that while the per-step computational cost of the full ForceFM (baseline + force model) is approximately **1.3× that of the baseline model alone**, this additional cost is more than offset by a significant reduction in the number of sampling steps required to achieve high-quality results.
>
> Looking ahead, we also envision opportunities to further optimize training and deployment. For instance, instead of training a separate force model, one could incorporate the guidance signal via **parameter-efficient fine-tuning techniques such as LoRA (Low-Rank Adaptation)** applied directly to the baseline model. This would drastically reduce training costs and eliminate any additional inference cost, while still enabling effective energy-based guidance.
>
> **Q4: Practical utility.**
>
> Yes, ForceFM shows strong potential for real-world application in drug discovery, particularly in structure-based virtual screening and hit refinement. **First, ForceFM is inherently compatible with widely adopted scoring functions such as Vina, Glide, and Gnina.** This design allows ForceFM to be seamlessly integrated into existing docking workflows as a plug-and-play module, requiring minimal changes to current pipelines. For example, it can be used downstream of coarse docking to refine ligand poses and improve their physical plausibility.
>
> **Second, the model’s force-guided generation mechanism consistently produces conformations that are not only low in energy but also chemically realistic.** These high-quality poses are more reliable for downstream prioritization, such as rescoring, clustering, or initiating free energy calculations, thereby enhancing the efficiency and confidence of lead identification.
>
> **Finally, ForceFM is computationally efficient.** Despite incorporating an additional guidance network, its inference time is significantly reduced—nearly halved compared to unguided diffusion models—making it feasible for mid-to-large scale virtual screening.
>
> Although our current paper focuses on benchmarking performance, we are actively collaborating with experimental researchers on the AI-driven design of coronavirus methyltransferase inhibitors. **In this real-world setting, ForceFM is integrated into both the hit finding and lead optimization stages, where it provides high-quality, energetically favorable binding poses that serve as input to alchemical free energy workflows (both ABFE and RBFE).** These physically plausible initial conformations accelerate FEP convergence and improve the reliability of binding affinity predictions.
>
> **More broadly, ForceFM serves as a critical bridge between generative molecular design and physics-based scoring.** In our current pipeline, ligand candidates generated by a deep generative model (e.g., language-based or diffusion-based de novo design) are passed through ForceFM for docking pose generation, ensuring that only physically meaningful and synthetically accessible molecules are prioritized for further evaluation. The system is further enhanced with active learning strategies that iteratively retrain the generative model based on feedback from docking, ForceFM-based refinement, and FEP predictions. This closed-loop integration highlights ForceFM’s role not only as a docking engine, but also as a structure-quality filter and physics-informed sampler that improves both the efficiency and scientific rigor of end-to-end AIDD workflows.

---

> > ### Comment · Reviewer_5rW8 · 2025-08-05
> >
> > I have read the author's response, and I'm grateful for the author's clarification. I will adjust my score to 4.

---

> > > ### Author Response · Authors · 2025-08-05
> > > **Acknowledgement of Review**
> > >
> > > Thank you very much for your positive evaluation and for raising your score—we sincerely appreciate your constructive feedback and the time you have dedicated to reviewing our manuscript.

---

### Official Review · Reviewer_y7e8 · 2025-07-02

**Clarity:** 2
**Significance:** 3
**Originality:** 3
**Rating:** 5
**Confidence:** 5

**Summary:**

The paper describes an approach to integrate energy functions into a flow matching model for molecular docking.  The energy function is introduced through a force network that does not require ground truth forces, just energies, for training. The evaluations show this approach results in better, more realistic docked poses. The approach is effective across multiple energy functions and model architectures. Claims to generalizability are problematic, however.

**Questions:**

How does the method perform in a more realistic cross-docking scenario?

How sensitive is the model to the initial pocket definition?

Is the graph structure updated with each iterative update (since the distances between nodes will change)?

Why does table 2 say it is for flexible docking? Nothing in the text indications flexible docking (moveable receptor) is being performed.

The importance of integrating the force network into training isn't evaluated.  If one simply performed minimization on the generated poses using the evaluated tools, would you see similar improvements in accuracy?

Vina does not measure the chemical stability of a conformer, only of its interaction - please be careful with your language.

**Ethical Concerns:**

["NO or VERY MINOR ethics concerns only"]

**Final Justification:**

With the additional evaluations presented during rebuttal, the paper now makes a compelling case for the utility of the method for molecular tasks.  They make a compelling argument that their model generalizes and is effective.

**Limitations:**

yes

**Quality:**

2

**Strengths And Weaknesses:**

Strengths.  This is a good idea. The force network provides a general approach for integrating an arbitrary energy function.  The evaluations convincingly show that including force matching is beneficial. If it weren't for the unsupported claims about generalizability, this would be a solid paper.

The effect of force guidance on the number of sampling steps needs is an interesting observation.


Weaknesses.  The claims to generalizability are overstated and the evaluations do not support the claims. It is true the approach generalizes to multiple energy functions and other generative models (DiffDock), but the paper claims they address "concerns about overfitting to specific protein or ligand types."  No practitioner of molecular docking would agree that time splits are a useful approach to measure  generalization or that simply filtering by Uniprot identifiers is sufficient to remove train/test information leakage.
Please review https://arxiv.org/html/2402.18396v1 for a more rigorous and reasonable approach for accurately assessing generalization.


The model does not actually perform blind docking, but instead uses P2Rank to identify pockets and then does pocket-based docking. A more sensible comparison would be to do pocket-based docking across the board to factor out the influence of P2Rank.

---

> ### Author Rebuttal · Authors · 2025-07-30
>
> We sincerely thank reviewer for your careful evaluation and constructive feedback. The insightful comments and thoughtful suggestions have greatly helped us improve the quality and clarity of this manuscript. We have carefully considered each comment and made corresponding revisions to address the concerns raised. We believe that the manuscript has been significantly strengthened through this process, and we are grateful for the time and expertise the reviewer have generously shared.
>
> **W1/Q1. Generalization claims overstated; time‑split and UniProt filtering are insufficient.**
>
> Thank you for your insightful feedback. We’ll carefully check the language. The statement like ‘concerns about overfitting to specific protein or ligand types’ will be revised as ‘improve the generalization ability of model’.
>
> For the generalization assessment, we agree that stronger OOD evaluation is desirable. To better assess the generalizability of our method in a realistic cross-docking scenario, **we evaluate our model—trained solely on the PDBBind dataset—on the DockGen test set.** This setup provides a test of cross-domain generalization. The results are shown as follows. Our results show that when integrating our plug-and-play force guidance module with the baseline model, there is a significant improvement compared to the baseline model on this challenging OOD benchmark. This demonstrates that our method not only enhances docking accuracy but also improves robustness to structural and functional shifts across the proteome.
>
> | Method | Mean RMSD | % RMSD < 2 | Mean Centroid Distance | % Centroid Distance < 2 |
> | --- | --- | --- | --- | --- |
> | FABind | 19.1 | 1.3 | 18.1 | 14.2 |
> | FABind+ | 18.5 | 1.5 | 16.9 | 16.1 |
> | DiffDock | 16.2 | 5.3 | 14.5 | 21.4 |
> | DiffDock + Vina | 12.5 | 7.1 | 10.9 | 25.7 |
> | DiffDock + Conf | 13.4 | 6.9 | 12.5 | 24.6 |
> | DiffDock + Gnina | 12.9 | 7.2 | 11.2 | 25.3 |
> | DiffDock + Glide  | 12.6 | 7.2 | 11.9 | 26.6 |
> | Ours | 14.5 | 6.5 | 14.1 | 23.2 |
> | Ours + Vina | 12.3 | **8.1** | 10.4 | 26.8 |
> | Ours + Conf | 12.5 | 7.9 | 11.6 | 25.4 |
> | Ours + Gnina | 13.1 | 7.6 | 12.1 | 26.9 |
> | Ours + Glide | **12.2** | **8.1** | **10.3** | **27.8** |
>
> **W2/Q2. Method is pocket‑based, hence not blind docking. Sensitivity to pocket definition.**
>
> We appreciate the reviewer’s insightful comment regarding the role of pocket prediction in our method. As noted in prior work [1, 2, 3], predicting binding pockets can significantly reduce the conformational search space and improve the efficiency and accuracy of docking models. In this work, we adopt **P2Rank** to estimate the binding site center and initialize the ligand around it—this aligns with standard practice in blind docking evaluation.
>
> To ensure a fair comparison and factor out the influence of P2Rank, we conducted additional experiments. Since **FABind [2]** and **FABind+ [3]** incorporate built-in pocket prediction modules, we added a **DiffDock + P2Rank** baseline for direct comparison. As shown in the table below, integrating P2Rank improves the mean RMSD and centroid distance significantly, with minimal change in the %RMSD < 2\AA metric—indicating more precise pose sampling near the true binding site, even if the top-ranked pose isn’t always within the strict 2\AA threshold.
>
> | Method | Mean RMSD | % RMSD < 2 | Mean Centroid Distance | % Centroid Distance < 2 |
> | --- | --- | --- | --- | --- |
> | DiffDock | 7.5 | 38.2 | 5.5 | 60.8 |
> | DiffDock + P2Rank | 5.4 | 37.6 | 3.5 | 67.3 |
>
> More importantly, we find that our method is **robust to small perturbations in the predicted pocket location**. To evaluate sensitivity, we injected Gaussian noise with increasing variance ($\sigma$ = 1 to 10 \AA) into the P2Rank-predicted pocket center before ligand initialization. As shown below, performance remains stable across all noise levels up to $\sigma$ = 5 \AA, with only minor degradation at σ = 10 \AA—comparable to results without P2Rank at all.
>
> | Method | Mean RMSD | % RMSD < 2 | Mean Centroid Distance | % Centroid Distance < 2 |
> | --- | --- | --- | --- | --- |
> | Ours | **4.2** | 41.1 | 3.1 | 71.7 |
> | $\sigma$=1 | **4.2** | 41.2 | **3.0** | **72.3** |
> | $\sigma$=2 | 4.5 | **41.7** | 3.1 | 71.4 |
> | $\sigma$=3 | 4.3 | 41.5 | 3.1 | 71.7 |
> | $\sigma$=5 | 4.4 | 41.5 | 3.2 | 70.9 |
> | $\sigma$=10 | 4.8 | 39.8 | 3.8 | 66.1 |
> | W/o P2Rank | 4.8 | 39.4 | 3.7 | 66.9 |
>
> These results suggest that our model does **not rely heavily on precise pocket localization**, likely due to the random initialization of ligand conformations and the iterative refinement process in the diffusion framework. Even when initialized far from the true site, the force guidance module helps steer the ligand toward plausible binding regions.
>
> **Q3. Is graph rebuilt after each update?**
>
> Yes; edges are recomputed every step to guarantee geometric consistency and they are used as features in the neural network.
>
> **Q4. Table 2 says “flexible” although receptor is rigid.**
>
> We appreciate the reviewer for highlighting this imprecision. This label was inadvertently adopted from the original reporting style in the FABind[2]/FABind+[3] papers, which used the term to describe their experimental setup. To avoid misinterpretation, we will revise the table header to **"PDBBind Docking Performance"**, which more accurately reflects our evaluation protocol.
>
> **Q5. Could post‑minimization deliver same gains?**
>
> Thank you for the insightful feedback. We fully agree that a critical question is whether the improvements from our force-guided framework stem from **genuine integration during generation** or could be replicated by **post-hoc energy minimization** of poses generated by a standard model. To this end, we conducted a additional ablation experiment, where we took the top-1 poses generated by both **DiffDock (**w or w/o force model**)** and **our baseline model ("Ours")**, and applied **energy minimization using AutoDock Vina's local optimizer** (without re-docking).  The results are summarized in the table below.
>
> It can be observed that:
>
> - **Energy minimization dramatically improves PoseBuster scores**, increasing them from ~14–24% to **over 40%**. This confirms that **poor physical plausibility is largely due to local geometric distortions** (e.g., clashes, bad bond angles) that minimization can fix.
> - **However, minimization has limited impact on RMSD** — it slightly reduces mean RMSD but does not significantly improve the percentage of poses below 2\AA. In some cases, RMSD even increases slightly, likely because the minimizer pulls the ligand away from the crystal pose while optimizing internal energy.
> - **Combining force-guided generation with minimization yields the best of both worlds**: **45.8% PoseBuster pass rate**, significantly outperforming any other combination.
>
> | Method | Mean RMSD | % RMSD < 2 | PoseBuster |
> | --- | --- | --- | --- |
> | DiffDock | 7.5 | 38.2 | 14.3 |
> | DiffDock + Mini | 7.0 | 37.9 | 40.3 |
> | DiffDock + Viina | 3.8 | 43.5 | 24.6 |
> | DiffDock + Vina + Mini | 3.9 | 42.8 | 42.5 |
> | Ours | 1.7 | 41.1 | 17.5 |
> | Ours+ Mini | 1.8 | 40.8 | 43.1 |
> | Ours+ Viina | 1.1 | 48.6 | 29.7 |
> | Ours+ Vina + Mini | **1.2** | **46.2** | **45.8** |
>
> **Q6. Vina does not measure internal strain—wording.**
>
> Thank you for pointing out this issue. You are absolutely right that Vina’s scoring function does not model ligand internal strain. Instead, it provides an empirical docking score that heuristically approximates protein–ligand binding affinity. To improve scientific precision, we will revise the original phrase “**chemical stability**” to “**empirical binding affinity score**,” and carefully review the manuscript to ensure all related terminology aligns with domain conventions.
>
> [1] Guo H, Liu S, Lou Y, et al. Diffdock-site: A novel paradigm for enhanced protein-ligand predictions through binding site identification[C]//NeurIPS 2023 Generative AI and Biology (GenBio) Workshop. 2023.
>
> [2] Pei Q, Gao K, Wu L, et al. Fabind: Fast and accurate protein-ligand binding[J]. Advances in Neural Information Processing Systems, 2023, 36: 55963-55980.
>
> [3] Gao K, Pei Q, Zhang G, et al. Fabind+: Enhancing molecular docking through improved pocket prediction and pose generation[C]//Proceedings of the 31st ACM SIGKDD Conference on Knowledge Discovery and Data Mining V. 1. 2025: 330-341.

---

> > ### Comment · Reviewer_y7e8 · 2025-08-01
> >
> > The additional experiments greatly strengthen the paper. I will be raising my score.
> >
> > You did not answer another reviewer's question about the gnina score. Gnina produces a pose score (CNNscore), affinity prediction (CNNaffinity) and a Vina score.  Which one was used for guidance?

---

> > > ### Author Response · Authors · 2025-08-01
> > > **Acknowledgement of Review**
> > >
> > > Thank you very much for your positive evaluation and for raising your score—we sincerely appreciate your constructive feedback and the time you have dedicated to reviewing our manuscript.
> > >
> > > Regarding the question about the gnina scoring function: in our study, we used the Vina-style scoring function implemented in gnina as score function.

---

> > > > ### Comment · Reviewer_y7e8 · 2025-08-01
> > > >
> > > > That's a little concerning - somehow using the exact same scoring function is getting different results (is there a difference in protonation preparation?). It also weakens the evaluation since if you were using one of the CNN scores you would be demonstrating the use of an entirely different modality of scoring function.

---

> > > > > ### Author Response · Authors · 2025-08-01
> > > > >
> > > > > As shown in the experimental results, the overall performance using Vina and GNINA is very similar. Indeed, while GNINA implements the same Vina-style scoring function as AutoDock Vina, we observed that the scores are not always identical when using the same protein and ligand file inputs. In practice, the results are very close, but small discrepancies—sometimes exceeding 0.1 kcal/mol—can occur. These differences may stem from subtle variations in implementation, numerical precision, or conformational sampling between the two tools.
> > > > >
> > > > > Furthermore, we apply random perturbations to the ligand conformations during training for local energy landscape estimation. This introduces slight variability in each run, and since each experiment is performed with randomization, the resulting poses and scores can vary slightly even under the same scoring function.
> > > > >
> > > > > We fully agree with your suggestion about exploring more advanced scoring modalities. As a follow-up, we plan to conduct additional experiments using GNINA’s CNN-based scoring functions (e.g., CNNscore, CNNaffinity) as well as hybrid scoring strategies. These experiments will further demonstrate the adaptability and effectiveness of our method under different scoring paradigms, including those leveraging deep learning.
> > > > >
> > > > > Thank you again for this insightful comment.

---

> > > > > > ### Author Response · Authors · 2025-08-02
> > > > > > **Additional experiment**
> > > > > >
> > > > > > We have conducted additional experiments on PDBBind testset using **CNNscore** and **CNNaffinity** as the guiding potentials in our force-guided framework. The results are summarized below. This confirms that our method is not only compatible with classical physics-based scoring but also benefits from data-driven, deep learning-based energy models that capture complex protein-ligand interaction patterns.
> > > > > >
> > > > > > | Method | Mean RMSD | RMSD < 2 | Mean Centroid Distance | Centroid Distance < 2 |
> > > > > > | --- | --- | --- | --- | --- |
> > > > > > | GNINA + Affinity | 3.1 | 47.5 | 2.4 | 74.3 |
> > > > > > | GNINA + CNNscore | 3.2 | 46.3 | 2.5 | 72.9 |
> > > > > > | GNINA + CNNaffinity | 3.1 | 48.6 | 2.3 | 75.8 |
> > > > > >
> > > > > > We would like to thank the reviewer again for your constructive feedback and the time to review our manuscript.

---

### Decision · Program_Chairs · 2025-09-17

**Decision:**

Accept (poster)

**Comment:**

This paper introduces ForceFM, a novel framework that integrates a force-guided module into flow matching generative models for protein-ligand docking.

ForceFM presents a compelling advance in structure-based drug discovery by bridging generative modeling with physics-informed guidance. While some methodological and presentation issues were raised, the authors responded constructively and convincingly. The paper is technically solid and well-motivated. All the reviewers are positive about the paper. I recommend acceptance.